# Physiological temperature drives TRPM4 ligand recognition and gating

Jinhong Hu[1], Sung Jin Park[1], Tyler Walter[1,2], Ian J. Orozco[1,3], Garrett O'Dea[1], Xinyu Ye[1], Juan Du[1✉] & Wei Lü[1✉]

Temperature profoundly affects macromolecular function, particularly in proteins with temperature sensitivity[1,2]. However, its impact is often overlooked in biophysical studies that are typically performed at non-physiological temperatures, potentially leading to inaccurate mechanistic and pharmacological insights. Here we demonstrate temperature-dependent changes in the structure and function of TRPM4, a temperature-sensitive $Ca^{2+}$-activated ion channel[3–7]. By studying TRPM4 prepared at physiological temperature using single-particle cryo-electron microscopy, we identified a 'warm' conformation that is distinct from those observed at lower temperatures. This conformation is driven by a temperature-dependent $Ca^{2+}$-binding site in the intracellular domain, and is essential for TRPM4 function in physiological contexts. We demonstrated that ligands, exemplified by decavanadate (a positive modulator)[8] and ATP (an inhibitor)[9], bind to different locations of TRPM4 at physiological temperatures than at lower temperatures[10,11], and that these sites have bona fide functional relevance. We elucidated the TRPM4 gating mechanism by capturing structural snapshots of its different functional states at physiological temperatures, revealing the channel opening that is not observed at lower temperatures. Our study provides an example of temperature-dependent ligand recognition and modulation of an ion channel, underscoring the importance of studying macromolecules at physiological temperatures. It also provides a potential molecular framework for deciphering how thermosensitive TRPM channels perceive temperature changes.

Temperature sensitivity is a defining feature of many macromolecules, affecting their function in physiology[12–16]. Studies of these macromolecules at the biophysical level are commonly conducted at subphysiological temperatures to preserve protein integrity. However, this practice may not accurately reflect their function in the human body—a consideration that is particularly crucial for thermosensing ion channels, such as members of the transient receptor potential (TRP) channels[17–27]. Among them, the TRPM4 channel, a member of the melastatin subfamily[4,28], has been identified by us as a notable example.

TRPM4 is widely expressed in various tissues and has important roles such as cellular depolarization, cardiac rhythm generation and immune response[29–32]. Mutations in TRPM4 are linked to serious cardiac conditions such as Brugada syndrome[32]. TRPM4 and its closest homologue TRPM5 are monovalent cation-selective channels; they are activated by intracellular $Ca^{2+}$ with a weak voltage dependence; and their activities are regulated by small molecules and lipids[5,7,8,33–36]. Like many TRP channels, they exhibit pronounced temperature sensitivity, with activities strongly intensifying near physiological temperatures[6,37]. However, the molecular mechanism underlying this temperature impact remains unclear, as existing structural data, from our group and others, have been limited to non-physiological temperature[10,11,38–40].

Here we investigated the structure, function and pharmacology of human TRPM4 channel at physiological temperature. Our findings revealed a striking phenomenon: TRPM4 can adopt distinct conformations at different temperatures, markedly influencing where and how ligands interact with them. Similar temperature-dependent structural changes and ligand recognition have not been reported in the structures of TRPV1 and TRPV3 activated by heat[41,42]. Thus, our findings and those of others suggest a complex and diverse thermal response across different protein families.

## Structure determination of TRPM4 at 37 °C

At room temperature (around 22 °C), cells overexpressing human TRPM4 exhibited relatively small $Ca^{2+}$-activated currents (Extended Data Fig. 1a). By contrast, near physiological temperature (about 37 °C), we observed a substantial increase in current magnitude (Extended Data Fig. 1a), consistent with a previous report that TRPM4 is a temperature-sensitive channel[6]. This observation underscores the distinct properties of TRPM4 at physiological temperatures versus room temperature, hinting that the differences may be even more pronounced compared with at the lower temperatures (4–18 °C) that are commonly used in cryo-electron microscopy (cryo-EM) sample preparation. This prompted us to investigate the molecular mechanism of TRPM4 under physiological temperatures. Purified human TRPM4 demonstrated high stability (Extended Data Fig. 2), providing a basis for its structural determination at physiological temperature.

[1]Van Andel Institute, Grand Rapids, MI, USA. [2]Present address: Zoetis, Kalamazoo, MI, USA. [3]Present address: AnaBios, San Diego, CA, USA. ✉e-mail: juan.du@vai.org; wei.lu@vai.org

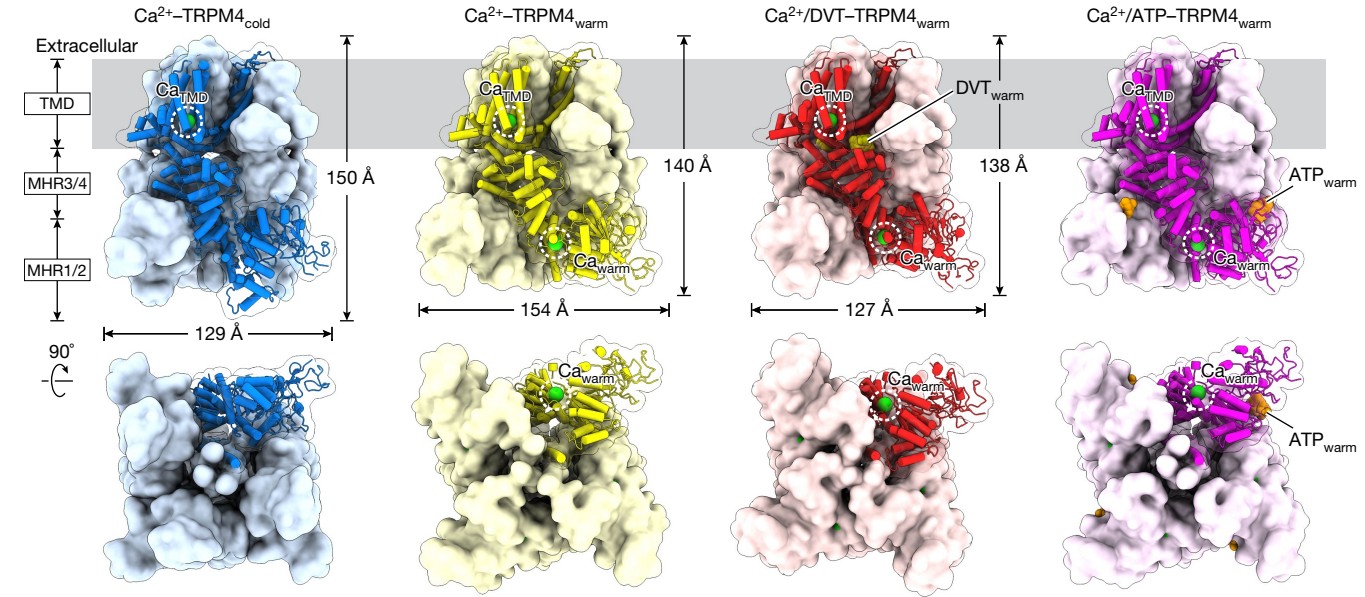

**Fig. 1 | The overall structures of TRPM4$_{cold}$ bound to Ca$^{2+}$ and TRPM4$_{warm}$ bound to Ca$^{2+}$, Ca$^{2+}$ and DVT, or Ca$^{2+}$ and ATP.** The structures are shown as a surface representation with one subunit in cartoon, viewed parallel to the membrane (top row) or from the intracellular side (bottom row).

We incubated TRPM4 with saturating Ca$^{2+}$ at 37 °C before grid preparation in a Vitrobot chamber, which was also maintained at 37 °C.

In the cryo-EM map, the transmembrane domain (TMD) is well resolved; however, the intracellular domain (ICD) containing the MHR1/2 and MHR3/4 domains showed a pronounced deterioration. In particular, the MHR1/2 domain appeared predominantly disordered with spike-like features (Extended Data Fig. 3), suggesting a mixture of different conformations. Indeed, three-dimensional (3D) classification at the single-subunit level revealed two classes with distinct ICD conformations (Extended Data Fig. 3). The first, representing around 25% of the particles, resembled the TRPM4 conformation observed at 4–18 °C (refs. 10,11,38–40). The second, representing around 75% of the particles, is a conformation in which the MHR1/2 domain is tilted towards the membrane compared with the first class (Extended Data Fig. 3). We refer to these conformations as 'cold' and 'warm', reflecting the temperatures at which they were identified and for ease of discussion. As discussed later, we speculate that both conformations may co-exist at physiological temperatures, with their equilibrium controlled by local Ca$^{2+}$ concentration.

Motivated by the identification of temperature-dependent conformations, we investigated whether ligand recognition and modulation of TRPM4 may also be affected by temperature. We focused on the ligands targeting the ICD given its pronounced conformational changes, including decavanadate (DVT), which positively modulates TRPM4 voltage dependence[8], and ATP, a TRPM4 inhibitor[9]. To this end, we performed cryo-EM studies on the TRPM4 samples prepared with saturating Ca$^{2+}$ at 37 °C in the presence of DVT and ATP, respectively (Extended Data Figs. 4 and 5). These conditions yielded both the cold and warm conformation, and we identified ligand-binding sites that are unique to the warm conformation. We also determined the TRPM4 structure in the presence of Ca$^{2+}$ at 18 °C, as well as in the presence of EDTA at 37 °C, both of which yielded exclusively the cold conformation (Extended Data Fig. 6a,b), indicating that neither Ca$^{2+}$ nor temperature alone shifts the cold-to-warm equilibrium. Given that all of the cold conformations in this study resembled previously published TRPM4 structures[10,11,38–40], our structural analysis and discussion will mainly address the warm conformation and its ligand-binding sites.

## Temperature shapes TRPM4 architecture

We determined human TRPM4 structures at physiological temperature in different functional states, including the apo state (EDTA–TRPM4), an agonist-bound state (Ca$^{2+}$–TRPM4$_{warm}$), an agonist- and positive-modulator-bound state (Ca$^{2+}$/DVT–TRPM4$_{warm}$) and an agonist- and inhibitor-bound state (Ca$^{2+}$/ATP–TRPM4$_{warm}$), at resolutions of 3.1–3.2 Å (Extended Data Tables 1 and 2 and Extended Data Fig. 7). All of the structures had a vertical arrangement of the TMD, MHR3/4 and MHR1/2 domains from top to bottom (Fig. 1). The warm conformation showed pronounced vertical compression and horizontal expansion of the ICD compared with the cold conformation (Fig. 1 (blue versus yellow)).

Three new ligand-binding sites were identified exclusive to the warm conformation. A Ca$^{2+}$-binding site, named Ca$_{warm}$, is located at the interface between the MHR1/2 and MHR3/4 domains (Fig. 1). A DVT-binding site is situated at the TMD and ICD interface (Fig. 1). This site, termed DVT$_{warm}$, is distinct from the two DVT-binding sites known in the cold conformation[10]. Lastly, an ATP-binding site, ATP$_{warm}$, is located between the MHR1/2 domain and the rib helix of the neighbouring subunit (Fig. 1), approximately 20 Å from the previously identified ATP$_{cold}$ site[11].

Looking into the TMD, the Ca$^{2+}$-bound cold and warm conformations have subtle but noticeable differences in the S1–S4 domain, S4–S5 linker and TRP helix due to the movement of the ICD that is in direct contact with these regions (Supplementary Video 1). However, their pore domains are nearly identical, each featuring a closed ion-conducting pore (Fig. 2). This implies that the combined effect of Ca$^{2+}$ and physiological temperature, although inducing large conformational changes in the ICD, is insufficient to open TRPM4 at 0 mV potential. This aligns with electrophysiological data demonstrating that Ca$^{2+}$-induced TRPM4 activation at 37 °C is outward rectifying and the currents at negative membrane potentials desensitize rapidly (Extended Data Fig. 1a,c,e). We therefore propose that the Ca$^{2+}$–TRPM4$_{warm}$ structure represents a pre-open state or desensitized state.

Notably, the addition of DVT resulted in an open pore in the Ca$^{2+}$/DVT–TRPM4$_{warm}$ structure with a radius of 2.5 Å (Fig. 2), allowing monovalent cations to pass through[4]. This open configuration is coherent with electrophysiological evidence that DVT renders TRPM4 largely voltage independent[8], but was not observed in the previously reported

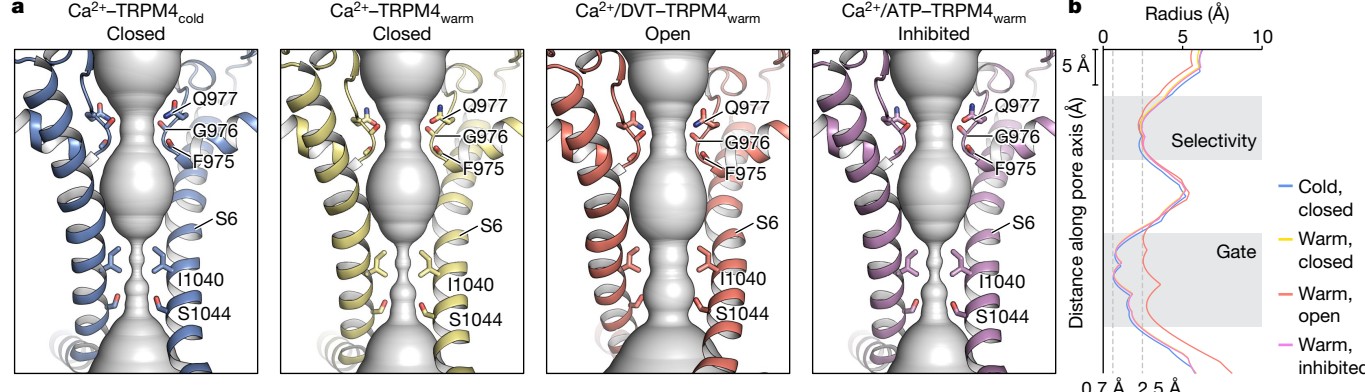

**Fig. 2 | The ion-conducting pore. a**, The profiles of the ion-conducting pore (shown as a surface representation) in different functional states, viewed parallel to the membrane. The pore region (shown as a cartoon) and residues (shown as sticks) forming the gate and the selectivity filter in two subunits are depicted. **b**, The pore radius along the pore axis.

structure in the DVT-bound cold conformation[10]. Thus, our finding reveals an open state of TRPM4, highlighting the intricate interplay between temperature, ligand binding and channel gating. By contrast, the ATP-bound warm conformation yielded a closed pore, representing an inhibited state (Fig. 2).

## TRPM4 conformational dynamics

To elucidate the molecular basis of the conformational dynamics between cold and warm conformations, we compared the structures of $Ca^{2+}$–$TRPM4_{warm}$ and $Ca^{2+}$–$TRPM4_{cold}$, and observed marked structural changes throughout the protein. This conformational disparity is especially evident in the MHR1/2 domains, which, when superimposing a warm subunit with a cold subunit using the MHR3/4 domains, showed a pronounced rigid-body rotation towards the MHR3/4 domain (Fig. 3a (yellow versus blue) and Supplementary Video 1). This movement converged specific residues from the MHR1/2 and MHR3/4 domains, creating the $Ca_{warm}$ site (Fig. 3b). Furthermore, our analysis revealed that the TRPM4 warm conformation closely resembled the $Ca^{2+}$-bound zebrafish TRPM5 structure (Fig. 3a (yellow versus grey)), with the $Ca_{warm}$ site in TRPM4 being similar to the Ca ICD site in TRPM5[43], although the latter does not depend on physiological temperature.

To investigate the role of $Ca_{warm}$ in TRPM4 channel function, we mutated a key residue in the binding site, Glu396, to alanine and performed electrophysiological studies. The E396A mutant displayed two strong phenotypes compared with the wild type: the loss of the temperature-induced potentiation and the inability to activate at negative membrane potentials (Fig. 3d,e and Extended Data Fig. 1b,d,f). The crucial role of Glu396 in coordinating $Ca_{warm}$ led us to propose that these phenotypes stem from disrupted $Ca^{2+}$ binding at this site. To test this idea, we determined the structure of TRPM4(E396A) at 37 °C in the presence of saturating $Ca^{2+}$, yielding exclusively the cold conformation (Fig. 3a and Extended Data Fig. 6c). Accordingly, the $Ca_{warm}$ site was absent, despite a pronounced $Ca^{2+}$ signal at the Ca TMD site, suggesting that the two $Ca^{2+}$-binding sites are not allosterically coupled. This diverges from zebrafish TRPM5, of which abolishing $Ca^{2+}$ binding at the Ca ICD site markedly reduced the Ca TMD binding[43].

Together, our data demonstrate that the $Ca_{warm}$ site and physiological temperature cooperatively have an indispensable role in cold-to-warm transition of TRPM4, with the warm conformation being critical for TRPM4 activation under physiological conditions (Fig. 3d). This is particularly relevant as TRPM4 is expressed in various non-excitable cells with physiological membrane potentials that are normally negative. Even in excitable cardiomyocytes, TRPM4 contributes to cardiac electrical activity through its activation at negative membrane potentials[44].

## Temperature determines binding of DVT

At physiological temperature, although the agonist $Ca^{2+}$ triggered substantial structural rearrangements in the ICD of TRPM4, the ion-conducting pore remained closed. This aligns with the fact that TRPM4 currents became outwardly rectifying shortly after initial exposure to intracellular $Ca^{2+}$, indicating a low open probability at zero or negative potentials (Fig. 3d,e). Our previous effort to capture an open-state structure at low temperatures using DVT was unsuccessful[10], despite the identification of two DVT-binding sites, $DVT_{cold1}$ and $DVT_{cold2}$ (Fig. 4a). This earlier discrepancy between structural and functional data now makes sense in light of our new data showing that mutations at these sites did not affect the modulatory effect of DVT (Fig. 4a and Extended Data Fig. 8a–i). This observation suggests that these sites are functionally irrelevant, and that DVT may bind non-specifically to negatively charged cavities.

Prompted by the identification of the warm conformation, we revisited this discrepancy by determining TRPM4 with $Ca^{2+}$ and DVT at 37 °C, revealing both cold and warm conformations (Extended Data Fig. 4). While the cold conformation retained the two $DVT_{cold}$-binding sites, the warm conformation featured a robust DVT density in a different cavity at the TMD–ICD interface, termed $DVT_{warm}$ (Fig. 4b). To assess the functional relevance of the $DVT_{warm}$ site, we neutralized the charge on each of the seven positively charged residues and performed electrophysiological experiments. Notably, apart from K928A, which did not generate $Ca^{2+}$-activated currents, six out of the remaining seven mutants became insensitive to DVT's voltage modulation effect (Fig. 4b and Extended Data Fig. 8a,j–p). Our data therefore support that the $DVT_{warm}$ site is the relevant site for voltage modulation. This $DVT_{warm}$ cavity, which is present in all TRPM channels, may represent a universal drug site for modulating TRPM family channels.

To understand how temperature allocates DVT into different sites in TRPM4, we analysed the electrostatic surface potential of the cold and warm conformations, considering that DVT is strongly negatively charged (Extended Data Fig. 9). We found that the structural rearrangement of the ICD from cold to warm reversed the electrostatic surface potential of the $DVT_{cold}$ sites from positive to negative, effectively preventing DVT from binding to these sites. On the other hand, the $DVT_{warm}$ cavity remained positively charged in both conformations. However, it became compatible with DVT binding only when the cavity is narrowed—probably to better accommodate the size of a DVT molecule—as MHR3/4 moved towards the TMD after the cold-to-warm transition.

## TRPM4 channel opening at 37 °C

To elucidate the activation mechanism of TRPM4 at physiological temperatures, we compared the structural differences between

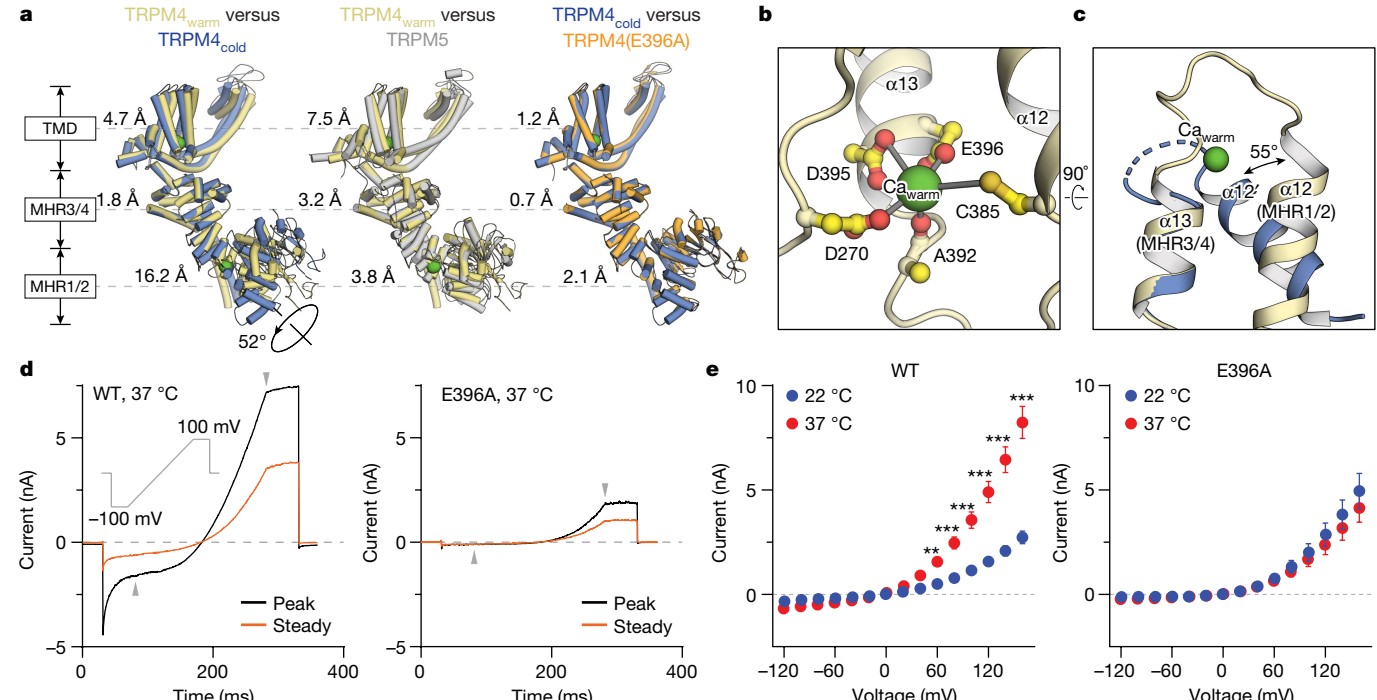

**Fig. 3 | Temperature and Ca²⁺ govern the cold-to-warm transition.**

**a**, Comparisons between subunits of Ca²⁺-bound wild-type TRPM4 cold and warm conformations, Ca²⁺-bound TRPM4(E396A) and Ca²⁺-bound zebrafish TRPM5 (Protein Data Bank (PDB): 7MBQ), superimposed using MHR3/4 (residues 391–687 in TRPM4 and 332–627 in TRPM5). The root mean squared deviation is shown for domains. **b**, The Ca_warm site; interactions between Ca²⁺ and coordinating residues are indicated by grey lines. **c**, Comparison of the Ca_warm site in the TRPM4 cold (blue) and warm (yellow) conformations by aligning helix α13 (residues 374–382), which forms half of the site. The angle between α12, which forms the other half of the site, is indicated by the black arrow. **d**, Whole-cell voltage-clamped currents were measured using patch pipettes containing 1 µM free calcium in tsA cells overexpressing wild-type TRPM4 (left) and the TRPM4(E396A) variant (right) at 37 °C. A protocol was applied every 5 s to monitor current changes, initiating at −100 mV for 50 ms,

ramping to +100 mV over 200 ms, then maintaining +100 mV for 50 ms; the holding potential was 0 mV. The black and orange traces represent average peak and steady-state current traces ($n = 5$ each for WT and E396A). **e**, After reaching the steady-state current in **d**, additional measurements were made using a multistep voltage-clamp protocol from −120 mV to 160 mV. Representative traces are shown in Extended Data Fig. 1a,b. Current amplitudes at the end of each pulse are plotted as a function of clamp voltage. $n = 16$ (22 °C, WT), $n = 15$ (37 °C, WT), $n = 8$ (22 °C, E396A) and $n = 7$ (37 °C, E396A) particles. Data are mean ± s.e.m. Statistical analysis was performed using two-way analysis of variance (ANOVA) with Bonferroni's post hoc test; *$P < 0.05$, **$P < 0.01$, ***$P < 0.001$. The $P$ values for the 60 to 160 mV steps on the left are as follows: $P = 0.0015$, $P < 0.0001$, $P < 0.0001$, $P < 0.0001$, $P < 0.0001$ and $P < 0.0001$, respectively.

Ca²⁺–TRPM4_cold and Ca²⁺–TRPM4_warm (Fig. 4c), as well as between Ca²⁺–TRPM4_warm and Ca²⁺/DVT–TRPM4_warm (Fig. 4d). This analysis revealed that physiological temperature and Ca²⁺ synergistically triggered an upward swing of the ICD, creating the Ca²⁺_warm site and priming the DVT_warm site for binding (Fig. 4c). However, this movement was confined to the ICD, leaving the ICD–TMD interface and the TMD largely unchanged. This suggests the need for other factors, such as membrane potential or allosteric modulators, to facilitate channel opening, as demonstrated by DVT_warm binding.

DVT_warm, densely packed with six negative charges, is situated between the ICD and TMD near the cytoplasmic surface of the membrane, attracting the positively charged membrane-facing side of the MHR3/4 domain towards it. As a result, the entire ICD moved towards the TMD in a rigid-body manner, elevating the S1–S4 domain towards the extracellular side through the TRP helix that bridges the ICD to the TMD (Fig. 4d). Meanwhile, the pull from the four DVT_warm molecules on the positively charged TRP helix and S4–S5 linker caused the pore-lining S6 helix to move outwards, therefore opening the ion-conducting pore (Fig. 4e,f and Supplementary Video 1).

## Temperature dictates ATP binding

We next extended our studies to a third type of ligand, ATP, a known endogenous TRPM4 inhibitor that also binds to its ICD[9,11,33]. At room

temperature, ATP effectively inhibited the Ca²⁺-induced TRPM4 current with a half-maximum inhibitory concentration (IC₅₀) in the micromolar range[9,11,33] (Fig. 5a,b). This was puzzling because it implies that TRPM4 would be constantly inhibited under cellular conditions in which the concentration of cytosolic free ATP is an order of magnitude higher than the IC₅₀ (ref. 45). However, when we conducted electrophysiological experiments at 37 °C, we observed a substantial decrease in ATP's inhibitory effect, with less than 50% of the current inhibited even at millimolar concentrations (Fig. 5a,b). This result indicates that ATP is not a potent inhibitor under physiological conditions, emphasizing the important role of temperature in modulating the inhibitory potency of ATP.

Motivated by the identification of the DVT_warm site, we examined the possibility that ATP may also exhibit a temperature-dependent binding and inhibition mechanism. Cryo-EM analysis of TRPM4 prepared with saturating Ca²⁺ and ATP at 37 °C yielded both cold and warm conformations (Extended Data Fig. 5). Importantly, we observed that ATP indeed occupied different locations in these two conformations (Fig. 5c). In the cold conformation, ATP bound at the interface between the MHR1/2 domain and the adjacent MHR3/4 domain, consistent with the previous findings[11]. At this ATP_cold site, the adenosine moiety of ATP was tightly encapsulated in a cleft in the MHR1/2 domain, forming extensive hydrophilic and hydrophobic interactions with surrounding residues, while its triphosphate tail loosely engaged with a positively

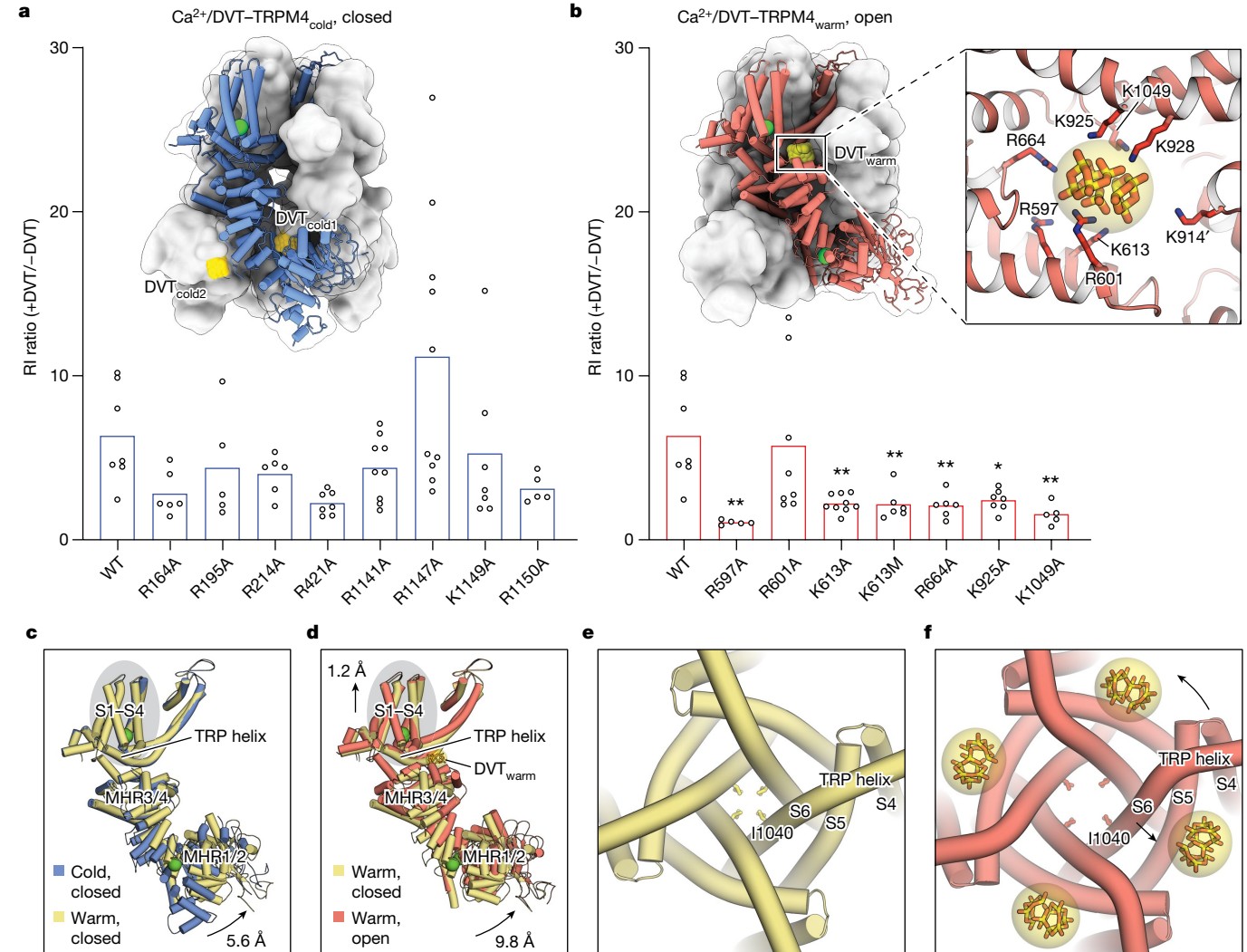

**Fig. 4 | Temperature determines DVT binding and modulation. a,b**, The rectification index (RI) ratio for TRPM4 wild type and mutants at the two DVT_cold sites (**a**) and mutants at the DVT_warm site (**b**). The RI is defined as the current ratio of $I(-120 \text{ mV})/I(+120 \text{ mV})$, and the RI ratio is defined as RI(+DVT)/RI(−DVT). Statistical analysis was performed using one-way ANOVA with Bonferroni's post hoc test (WT was compared with each mutant). The *P* values in **b** from left (R597A) to right are as follows: $P = 0.0013, P > 0.9999, P = 0.0042, P = 0.0100, P = 0.0085, P = 0.0122$ and $P = 0.0042$, respectively. Representative traces are provided in Extended Data Fig. 8. Insets: the structure of Ca²⁺/DVT–TRPM4_cold (**a**) and Ca²⁺/DVT–TRPM4_warm (**b**) as a surface representation, with one subunit as a cartoon and DVT molecules as yellow spheres. The magnified view in **b** shows the interactions within the DVT_warm site, which is encompassed by the MHR3/4 domain, pore domain and TRP helix of one subunit, along with the S4–S5 linker of the adjacent subunit. The DVT molecule is shown as sticks with a transparent surface, and the surrounding positively charged residues are shown as sticks. **c,d**, Comparisons of Ca²⁺-bound cold (blue) versus warm (yellow) conformations (**c**), and Ca²⁺-bound warm versus Ca²⁺/DVT-bound warm conformations (**d**), by aligning the tetrameric pore helix and loop (residues 958–989). A single subunit is depicted, with the centre-of-mass movement of the MHR1/2 domain indicated to represent the motion of the ICD. The centre-of-mass movement of the S1–S4 movement is also indicated. **e,f**, The pore domain in the Ca²⁺-bound warm (**e**) and Ca²⁺/DVT-bound warm (**f**) conformations viewed from the intracellular side. The movements of the S6 helix, TRP helix and S4–S5 linker caused by DVT binding are indicated. The side chain of Ile1040, which forms the channel gate, is shown as sticks. The DVT molecule is shown as sticks with a transparent surface.

charged cap (Arg421) on the adjacent MHR3/4 domain[11]. The transition to the warm conformation shifted the MHR1/2 domain toward the TMD, moving the 'adenosine cleft' to near the N-terminal tip of the rib helix. Notably, this movement also transferred ATP within the cleft to this new location, ATP_warm (Fig. 5c). Here, while the interactions of the adenosine group within the MHR1/2 domain remained essentially unchanged, the triphosphate tail became surrounded by positively charged residues from the rib helix (Fig. 5d).

Comparing the structures of the non-conducting warm conformation with and without ATP revealed that ATP_warm binding did not induce major conformational changes. Instead, it appears that ATP binding locked the channel in a non-conducting warm conformation by impeding the ICD movement necessary for channel activation (Fig. 4d). To test

this hypothesis, we determined the structure of TRPM4 with Ca²⁺, DVT and ATP at 37 °C (Extended Data Fig. 6d). Indeed, this structure closely resembled the ATP-bound non-conducting warm conformation, but not the DVT-bound warm open conformation, with a weak 'dust-like' density at the DVT_warm site and a closed pore (Extended Data Fig. 6e–g). We speculate that ATP_cold similarly inhibits the channel at lower temperature by locking it in a non-conducting cold conformation. Furthermore, it is possible that ATP_cold binding has a higher affinity compared with ATP_warm, which may explain ATP's increased inhibitory potency at room temperature. Together, our data suggest that, although the adenosine cleft has a critical role in ATP recognition, the mechanism of ATP inhibition is determined by its spatial location, which is temperature dependent.

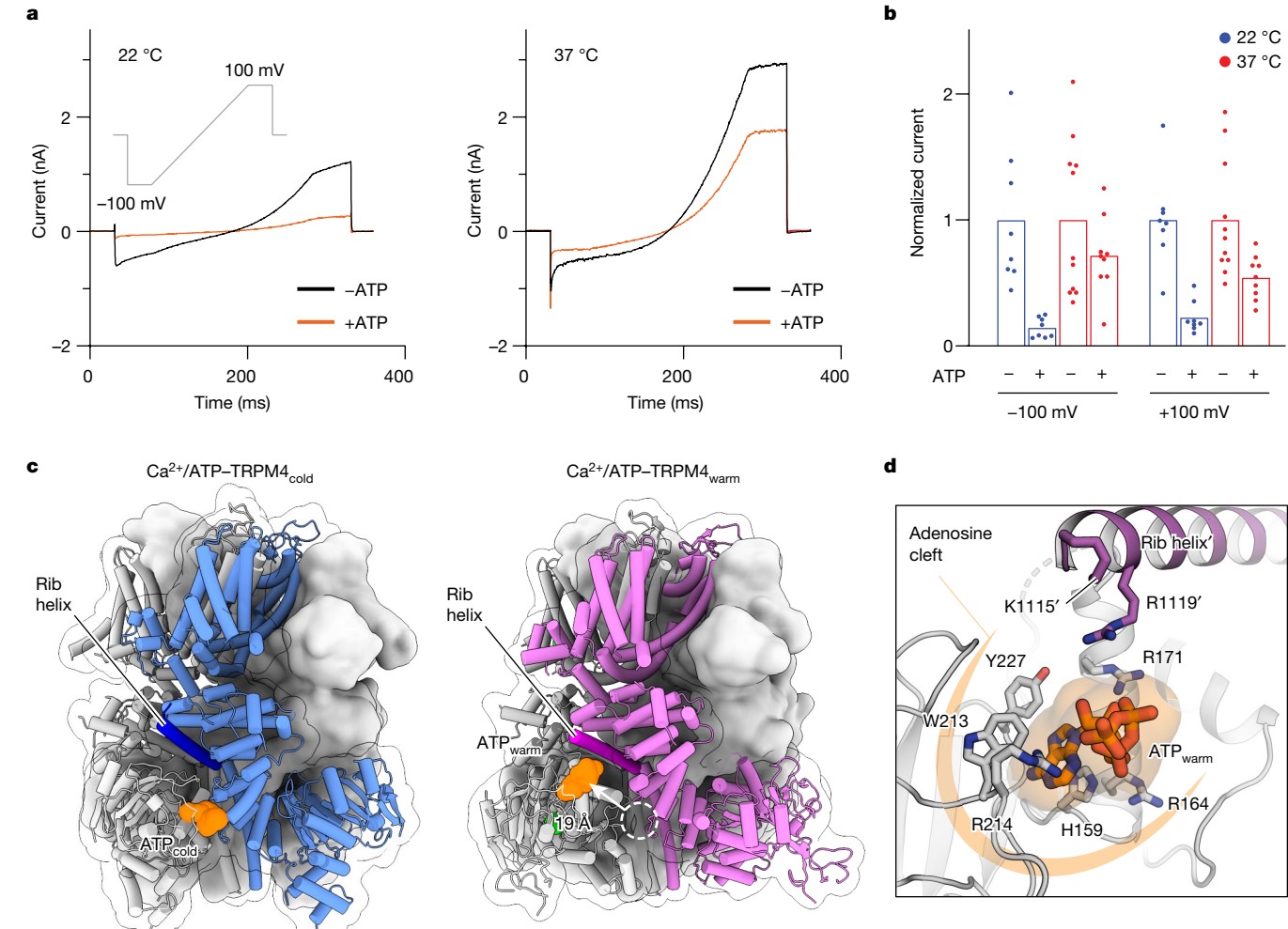

**Fig. 5 | Temperature dictates inhibitor binding and action. a**, Whole-cell currents were measured in tsA cells overexpressing wild-type TRPM4 at 22 °C (left) and 37 °C (right). A protocol was applied every 5 s to monitor current changes, initiating at −100 mV for 50 ms, then ramping to +100 mV over 200 ms, and finally holding at +100 mV for 50 ms. The black and orange traces represent the average steady-state current traces measured with 1 μM free $Ca^{2+}$ and 1 μM free $Ca^{2+}$ plus 5 mM ATP, respectively, in the pipette solution. $n = 8$ (−ATP, 22 °C), $n = 8$ (+ATP, 22 °C), $n = 11$ (−ATP, 37 °C) and $n = 9$ (+ATP, 37 °C). **b**, Normalized current amplitudes in the presence and absence of ATP (at 50 ms of holding potentials of +100 and −100 mV) of the experiments in **a** were plotted. Each point represents a single cell and the bars represent the mean. **c**, The structure of $Ca^{2+}$/ATP–TRPM4$_{cold}$ (left) and $Ca^{2+}$/ATP–TRPM4$_{warm}$ (right) as a surface representation, with one subunit as a cartoon and ATP molecules as orange spheres. The dashed white circle marks the position of the ATP$_{cold}$ site hypothetically mapped in the $Ca^{2+}$/ATP–TRPM4$_{warm}$ structure, with its distance to the ATP$_{warm}$ site indicated. **d**, A magnified view of the interactions within the ATP$_{warm}$ site. One subunit is coloured grey, while the adjacent subunit is coloured magenta. The ATP molecule is shown as sticks with a transparent surface, and the surrounding residues are shown as sticks.

## Discussion and conclusion

Our study of the temperature-sensitive TRPM4 channel at physiological temperatures offers critical insights into its temperature-dependent structural dynamics, ligand recognition and gating mechanisms. A key finding is the warm conformation, which differs from the cold conformation observed at lower, non-physiological temperatures. Our analysis—although currently limited to the biophysical level—suggests that both cold and warm conformations coexist in physiological conditions, with their equilibrium modulated by the local $Ca^{2+}$ concentration. The identification of the open-state structure of TRPM4 at physiological temperature underscores the importance of the cold-to-warm transition in channel activation. However, the precise interplay of temperature and $Ca^{2+}$ during this transition remains to be further explored. A plausible explanation is that elevated temperatures may weaken the intersubunit interface in the cold conformation[46,47], potentially facilitating, in conjunction with $Ca_{warm}$ binding, the transition to the warm conformation with new intersubunit

interfaces. The cold-to-warm transition of TRPM4 also profoundly affects its ligand recognition, highlighted by the identification of temperature-dependent binding for ligands across three principal categories: $Ca^{2+}$, the endogenous agonist; DVT, a positive modulator; and ATP, an inhibitor. Importantly, we provide compelling evidence that these sites—in contrast to those observed or absent in the cold conformation—are functionally relevant.

It is important to acknowledge that discrepancies may arise between the intended and actual protein temperatures due to the limitations of the cryo-EM sample-freezing process. Indeed, the conformational dynamics captured in the cryo-EM data are likely to reflect a temperature lower than 37 °C at which the sample was prepared. We therefore propose that the warm conformation may exist even below 37 °C. This notion is supported, in part, by our electrophysiological studies on DVT using an inside-out configuration at room temperature (as conducting these experiments at 37 °C posed practical challenges, possibly due to the fragility of excised membrane at elevated temperatures). In these experiments, we were able to discern the phenotypes of mutations at

the DVT$_{warm}$ site, suggesting that the warm conformation is present in the cellular environment at room temperature.

In conclusion, our study elucidates the temperature-dependent mechanisms of ligand recognition and gating of TRPM4 and, more broadly, underscores the importance of considering temperature as a pivotal factor in both mechanistic studies and drug development. Moreover, while a direct role of TRPM4 in temperature sensing is yet to be established, its marked structural similarity to well-known thermosensors in the same family, including the heat sensor TRPM3 and the cool senor TRPM8[27,48–50], hints at a universal molecular framework for how these proteins sense temperature changes.

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

## Methods

### Human TRPM4 protein expression and purification

The gene encoding human full-length TRPM4 (UniProtKB: Q8TD43) was subcloned into pEG BacMam vector with a 2×Strep tag, GFP and a thrombin-cleavage site at the N terminus[51]. Bacmid and baculovirus of TRPM4 in a BacMam vector were generated, and P2 viruses were used to infect a suspension of tsA cells. Cells were incubated at 37 °C for 12 h. Subsequently, 10 mM sodium butyrate was added to the culture and the temperature was changed to 30 °C. The cells were collected 72 h after infection and resuspended in a buffer containing 150 mM NaCl and 20 mM Tris pH 8.0 (TBS buffer) in the presence of 1 mM phenylmethylsulphonyl fluoride, 0.8 μM aprotinin, 2 μg ml$^{-1}$ leupeptin and 2 mM pepstatin A. The cells were lysed by sonication. The membrane fraction was collected by centrifugation at 186,000$g$ using a 45 Ti rotor (Beckman Coulter) for 1 h at 4 °C. It was then homogenized with a Dounce homogenizer in TBS buffer supplemented with protease inhibitors. The protein was extracted from the membrane with TBS buffer supplemented with 1% GDN and protease inhibitors for 3 h at 4 °C. The solubilized proteins were loaded to Strep-Tactin resin. After washing with TBS buffer supplemented with 0.02% GDN, TRPM4 was eluted with the same buffer, supplemented with 10 mM desthiobiotin. The proteins were concentrated and further purified by size-exclusion chromatography (Superose 6). The peak fractions containing the channel were pooled and concentrated to 6 mg ml$^{-1}$.

### EM sample preparation and data acquisition

For sample preparation at 37 °C, purified TRPM4 was incubated with 5 mM calcium chloride or 5 mM EDTA, as required for each experiment, for 10 s. For samples requiring DVT or ATP, TRPM4 mixed with 5 mM calcium chloride was incubated at 37 °C for 30 s before the addition of DVT or ATP to final concentrations of 2 or 5 mM, respectively, followed by further incubation for 2 min. In experiments including both DVT and ATP, TRPM4 was first incubated with 5 mM calcium chloride at 37 °C for 30 s, then with 1 mM DVT for 2 min, and finally with 2 mM ATP for an additional 2 min. For all conditions, 2.5 μl of the 6 mg ml$^{-1}$ sample was applied to a glow-discharged Quantifoil holey carbon grid (gold, 2/1 μm size/hole space, 300 mesh). The grids were blotted for 1.5 s in the Vitrobot Mark III set to 100% humidity and 37 °C with a 15 s wait time before being plunge-frozen into liquid ethane cooled by liquid nitrogen.

For the Ca$^{2+}$-containing sample at 18 °C, purified TRPM4 was incubated with 5 mM calcium chloride at 18 °C for 10 s. After incubation, 2.5 μl of the 6 mg ml$^{-1}$ sample was applied to a glow-discharged Quantifoil holey carbon grid (gold, 2/1 μm size/hole space, 300 mesh). The grids were then blotted for 1.5 s in the Vitrobot Mark III set to 100% humidity and 18 °C, with a 15 s wait time, before being plunge-frozen into liquid ethane cooled by liquid nitrogen.

Images for all samples, except for those detailed below, were recorded using the FEI Titan Krios electron microscope at 300 kV and a nominal magnification of 105,000×. Data were collected on the Gatan K3 Summit direct electron detector in super-resolution mode, resulting in a binned pixel size of 0.828 Å, using SerialEM for automated acquisition[52]. Nominal defocus ranged from −1.2 to −1.9 μm.

For the Ca$^{2+}$-containing sample prepared at 18 °C, images were recorded on an FEI Arctica electron microscope at 200 kV and a nominal magnification of ×45,000. A Gatan K2 Summit direct electron detector in super-resolution mode was used, yielding a binned pixel size of 1.16 Å, with SerialEM for automated acquisition. Nominal defocus ranged from −1.4 to −2.2 μm.

For the Ca$^{2+}$/DVT/ATP-containing sample prepared at 37 °C, images were recorded on the FEI Arctica electron microscope at 200 kV and a nominal magnification of ×45,000. A Gatan K2 Summit direct electron detector in super-resolution mode was used, yielding a binned pixel size

of 0.92 Å, with SerialEM for automated acquisition. Nominal defocus ranged from −1.4 to −2.2 μm.

### Cryo-electron microscopy data analysis procedure

The detailed workflow for the data-processing procedure is summarized in Extended Data Figs. 3–7. In general, the raw super-resolution .tif video files for each dataset were motion-corrected and 2× binned using MotionCor2 (v.1.1.0)[53]. The per-micrograph defocus values were estimated using ctffind (v.4.1.10)[54]. Particle picking was performed using gautomatch (v.0.56) (https://github.com/JackZhang-Lab/Gautmatch) or topaz (v.0.2.4)[55] or RELION's template picking[56]. Junk particles were removed by rounds of 3D heterogeneous refinement using CryoSPARC[57]. Good particles were selected for homogeneous refinement with $C_4$ symmetry in CryoSPARC to generate a 3D map. Multiple rounds of CTF refinement and Bayesian polishing were performed in RELION to further improve the map resolution.

For datasets showing noticeable conformational heterogeneity in the ICD, symmetry expansion at the single-subunit level was done from the map refined with $C_4$ symmetry, followed by subtraction of the ICD. The subtracted images of the ICD underwent a round of local refinement with $C_1$ symmetry followed by 3D classification without image alignment in RELION, yielding two distinct conformations—warm and cold. A similar procedure was used to further improve the local resolution of the MHR1/2 domain. The best parts of all focused maps were combined in phenix[58] and used for model building. Homotetrameric particles of the warm conformation were identified and refined to generate the final tetrameric map.

For the Ca$^{2+}$/DVT/37 °C dataset, symmetry expansion at the single subunit level was done from the map refined with $C_4$ symmetry, followed by monomer subtraction. The subtracted images of the monomer underwent a round of local refinement with $C_1$ symmetry followed by 3D classification without image alignment in RELION. Classes that showed an open-pore conformation with strong DVT density were selected. Homotetrameric particles of this conformation were identified and refined to generate a final tetrameric map.

Map resolution estimates were based on the gold standard Fourier shell correlation 0.143 criterion for all datasets[56].

### Model building

Atomic models were generated by rigid-body fitting of the TMD, MHR1/2 and MHR3/4 and C-terminal domains from a published human TRPM4 model (PDB: 5WP6) into the final cryo-EM maps. Ligands were fitted into the density through real-space refinement using COOT[59]. The CIF file of DVT was generated using eLBOW[60]. DVT molecules, lacking distinctive features, were aligned within the density so their long axes corresponded with those of the densities. The models were then manually adjusted in COOT and subjected to phenix.real_space_refine to improve the model metrics. The final models were validated using phenix.molprobity[61]. Figures were generated using PyMOL (Schrödinger LLC) and UCSF ChimeraX[62]. The profiles of the ion-conducting pore were calculated using HOLE[63].

### Electrophysiology

TsA201 cells expressing plasmids encoding N-terminal GFP-tagged human TRPM4 were used. Then, 1 day (for whole-cell recording) or 2 days (for inside-out recording) after transfection with plasmid DNA (100 ng μl$^{-1}$) and Lipofectamine 2000 (Invitrogen, 11668019), the cells were trypsinized and replated onto poly-L-lysine-coated (Sigma-Aldrich, P4707) glass coverslips. After cell attachment, the coverslip was transferred to a recording chamber. Whole-cell patch-clamp recordings were performed at room temperature (21–23 °C) or body temperature (36–38 °C). Signals were amplified using the Multiclamp 700B amplifier and digitized using the Digidata 1550B A/D converter (Molecular Devices). The whole-cell current was measured on the cells with an access resistance of less than 10 MΩ after the whole-cell configuration

was obtained. The whole-cell capacitance was compensated by the amplifier circuitry. A typical TRPM4 current shows a transient activation by intracellular $Ca^{2+}$ and subsequent desensitization to reach a steady state within 1–2 min (ref. 64). The ramp pulse from −100 to 100 mV for 200 ms with a holding potential of 0 mV was continuously applied to the cell membrane every 5 s to monitor the transient of the TRPM4 current. The step pulse from −120 to 160 mV for 200 ms was applied after steady state was reached. The inside-out patch was performed at room temperature (21–23 °C) to study the effect of DVT. A 200 ms step pulse from 160 mV to −120 mV (intracellular side relative to extracellular side) was applied. Electrical signals were digitized at 10 kHz and filtered at 2 kHz. Recordings were analysed using Clampfit v.11.3 (Axon Instruments), GraphPad Prism 10 and OriginPro 2024 (OriginLab). The standard bath solution contains 150 mM NaCl, 5 mM KCl, 1 mM $MgCl_2$, 2 mM $CaCl_2$, 12 mM mannitol, 10 mM HEPES, pH 7.4 with NaOH. For a whole-cell recording, the extracellular solution contains 150 mM NaCl, 10 mM HEPES, 1 mM $MgCl_2$ and 2 mM $CaCl_2$. The intracellular solution contains 150 mM NaCl, 10 mM HEPES, 5 mM EGTA, 4.45 mM $CaCl_2$ (1 μM of free $Ca^{2+}$). 5 mM di-sodium ATP is included in the intracellular solution described above to test the effect of ATP. In the inside-out patch mode, the extracellular solution (pipette) contains 150 mM NaCl, 10 mM HEPES, 1 mM $MgCl_2$ and 2 mM $CaCl_2$. The intracellular solution (bath) contains 150 mM NaCl, 10 mM HEPES, 5 mM EGTA. 5 mM EGTA was replaced by 5 mM $CaCl_2$ (or 5 mM $CaCl_2$ + 10 μM DVT).

### Reporting summary

Further information on research design is available in the Nature Portfolio Reporting Summary linked to this article.

### Data availability

Cryo-EM density maps have been deposited at the Electron Microscopy Data Bank under accession numbers EMD-44360 ($Ca^{2+}$–$TRPM4_{warm}$), EMD-44361 ($Ca^{2+}$–$TRPM4_{warm}$ subunit), EMD-44362 ($Ca^{2+}$/DVT–$TRPM4_{warm}$), EMD-44363 ($Ca^{2+}$/DVT–$TRPM4_{warm}$ subunit), EMD-44364 ($Ca^{2+}$/ATP–$TRPM4_{warm}$), EMD-44365 ($Ca^{2+}$/ATP–$TRPM4_{warm}$ subunit), EMD-44366 ($Ca^{2+}$–$TRPM4_{cold}$), EMD-44367 (EDTA–TRPM4), EMD-44368 ($Ca^{2+}$–TRPM4(E396A)) and EMD-44369 ($Ca^{2+}$/ATP–$TRPM4_{warm}$ + DVT). Structure models have been deposited at the RCSB PDB under accession codes 9B8W ($Ca^{2+}$–$TRPM4_{warm}$), 9B8X ($Ca^{2+}$–$TRPM4_{warm}$ subunit), 9B8Y ($Ca^{2+}$/DVT–$TRPM4_{warm}$), 9B8Z ($Ca^{2+}$/DVT–$TRPM4_{warm}$ subunit), 9B90 ($Ca^{2+}$/ATP–$TRPM4_{warm}$), 9B91 ($Ca^{2+}$/ATP–$TRPM4_{warm}$ subunit), 9B92 ($Ca^{2+}$–$TRPM4_{cold}$), 9B93 (EDTA–TRPM4) and 9B94 ($Ca^{2+}$–TRPM4(E396A)).

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

**Acknowledgements** We thank G. Zhao and X. Meng for the support with data collection at the David Van Andel Advanced Cryo-Electron Microscopy Suite; and the members of the high-performance computing team of VAI for computational support. W.L. is supported by National Institutes of Health (NIH) grants (R01HL153219 and R01NS112363). J.D. is supported by a McKnight Scholar Award, a Klingenstein-Simon Scholar Award, a Sloan Research Fellowship in neuroscience, a Pew Scholar in the Biomedical Sciences award and NIH grants (R01NS111031 and R01NS129804). J.H. is supported by American Heart Association postdoctoral fellowship (24POST1196982).

**Author contributions** W.L. and J.D. supervised the project. J.H. and T.W. performed protein purification, cryo-EM data collection and processing. G.O., X.Y. and J.H. generated the mutants. S.J.P. and I.J.O. performed electrophysiology experiments. All of the authors contributed to the manuscript preparation.

**Competing interests** The authors declare no competing interests.

### Additional information

**Correspondence and requests for materials** should be addressed to Juan Du or Wei Lü.

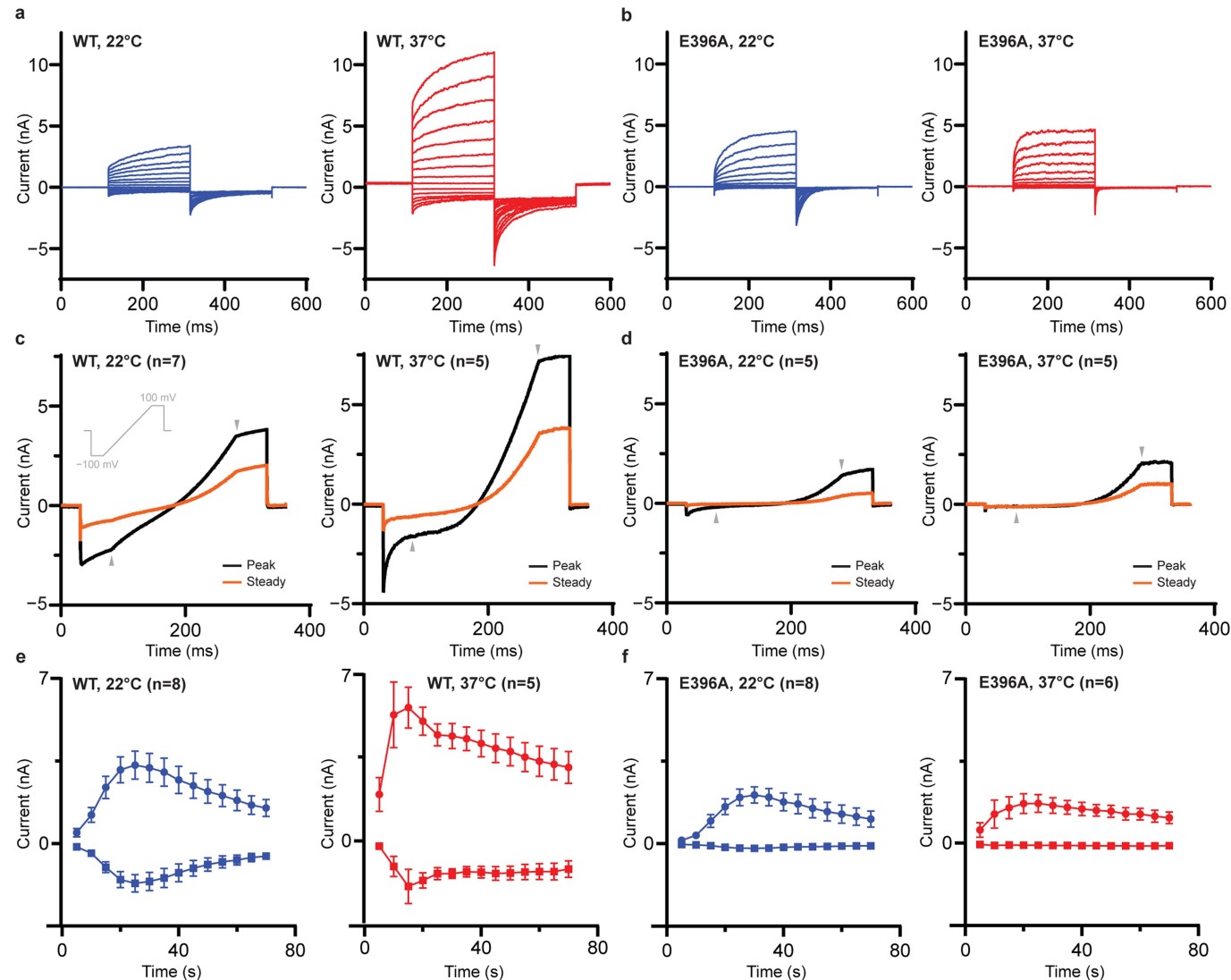

**Extended Data Fig. 1 | Comparison of thermosensitive currents of wild-type TRPM4 and its E396A variant. a,b,** Steady-state whole-cell voltage-clamped currents were measured using patch pipettes containing 1 µM free calcium in tsA cells overexpressing wild-type TRPM4 (**a**) and the E396A variant (**b**) at 22 °C (blue) and 37 °C (red). Voltage clamps were imposed from −120 mV to 160 mV with a final tail pulse at −120 mV, with a holding potential of 0 mV. **c,d,** Whole-cell voltage-clamped currents were measured using patch pipettes containing 1 µM free calcium in tsA cells overexpressing wild-type TRPM4 (**c**) and the E396A variant (**d**) at 22 °C (left) and 37 °C (right). A protocol was applied every 5 s to monitor current changes, initiating at −100 mV for 50 ms, then ramping to +100 mV over 200 ms, and finally at +100 mV for 50 ms, with a holding potential of 0 mV. The black and orange traces represent average peak and steady-state current traces (n = 7 and n = 5 for WT at 22 °C and 37 °C, and n = 5 and n = 5 for E396A at 22 °C and 37 °C). **e,f,** Mean current amplitudes at the end of the +100 mV and −100 mV pulses of experiments in (**c,d**) were plotted as a function of time, showing the activation and desensitization of TRPM4. The number of patches is indicated in the parentheses after the construct name. Error bars represent SEM.

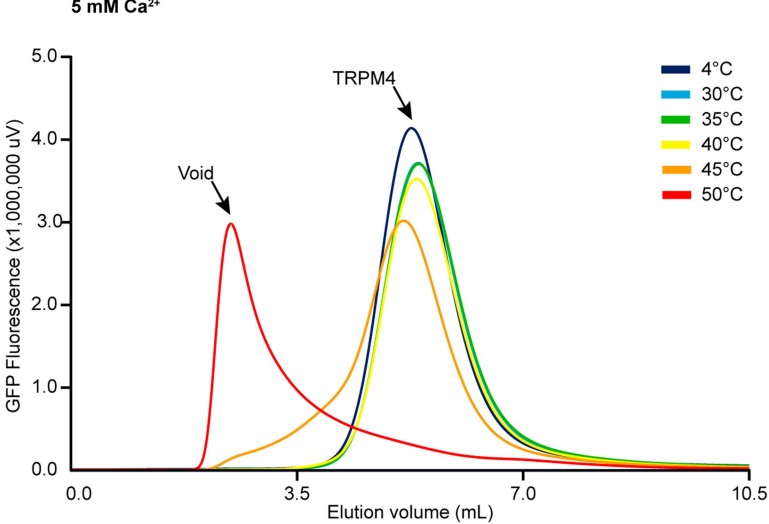

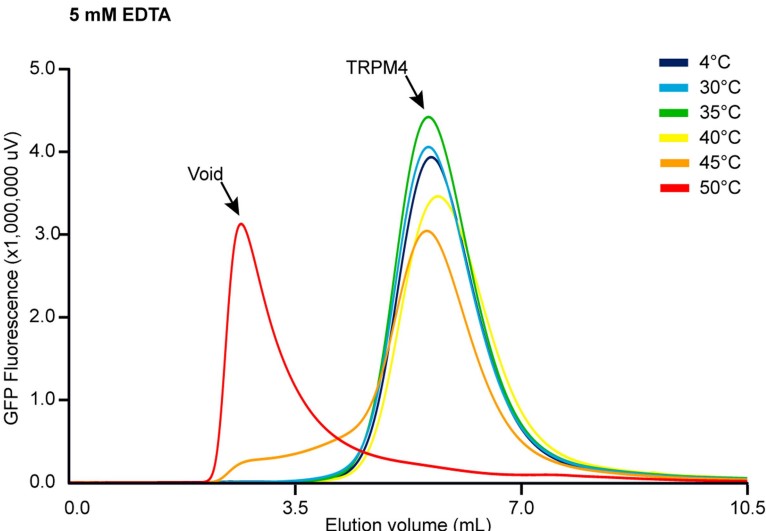

**Extended Data Fig. 2 | Thermostability test of human TRPM4 in the presence of 5 mM Ca²⁺ (top) or 5 mM EDTA (bottom) using fluorescence-detection size-exclusion chromatography.** Protein samples were incubated at various of temperatures for 10 min before loading onto a size-exclusion column.

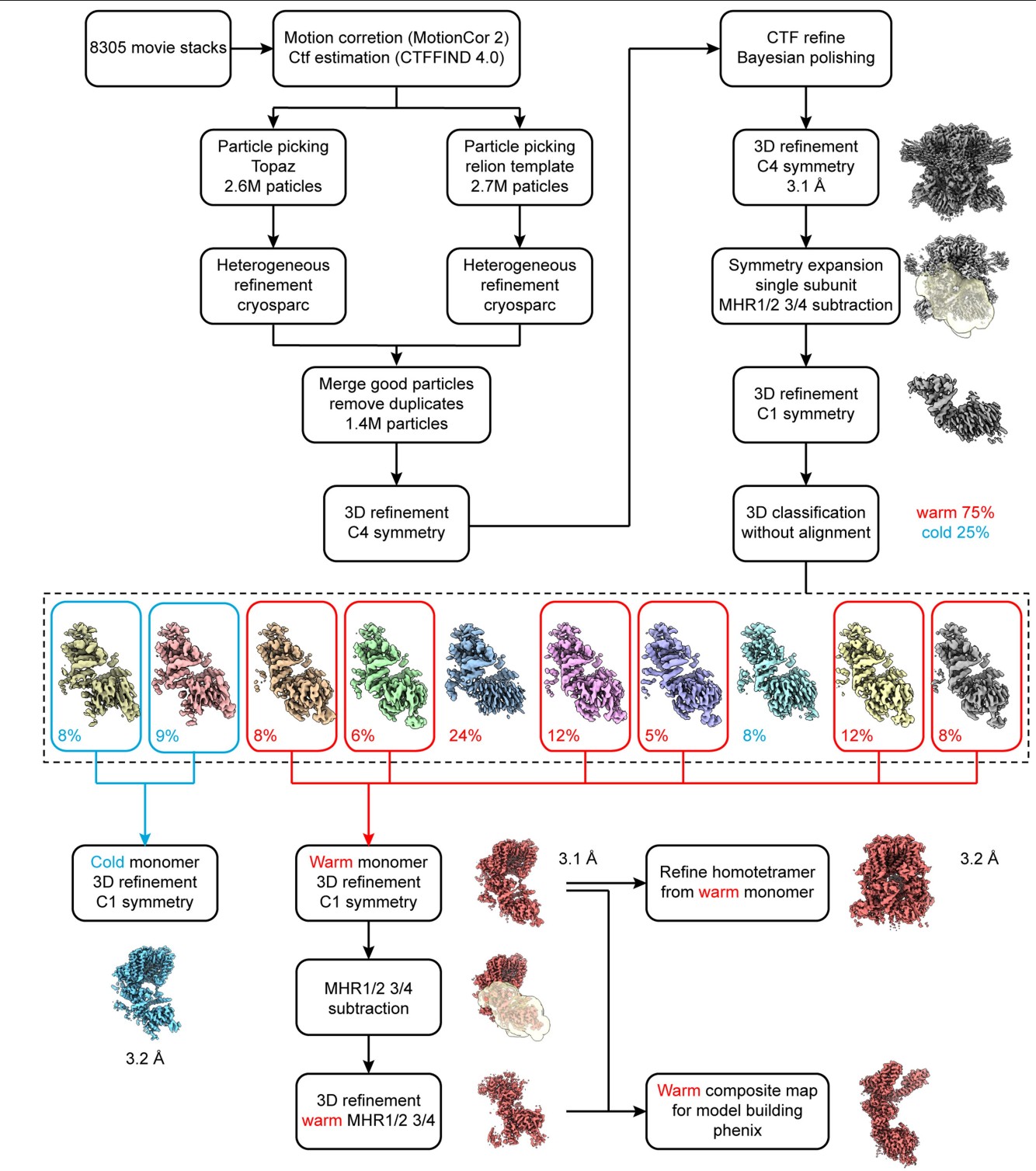

**Extended Data Fig. 3 | Cryo-EM data processing workflow of TRPM4 with 5 mM Ca²⁺ at 37 °C.** Key maps and the mask (shown as a transparent envelope) used to subtract the intracellular domain (MHR1/2/3/4) are included. This figure also shows the percentage of the particles in warm and cold conformations at the subunit level.

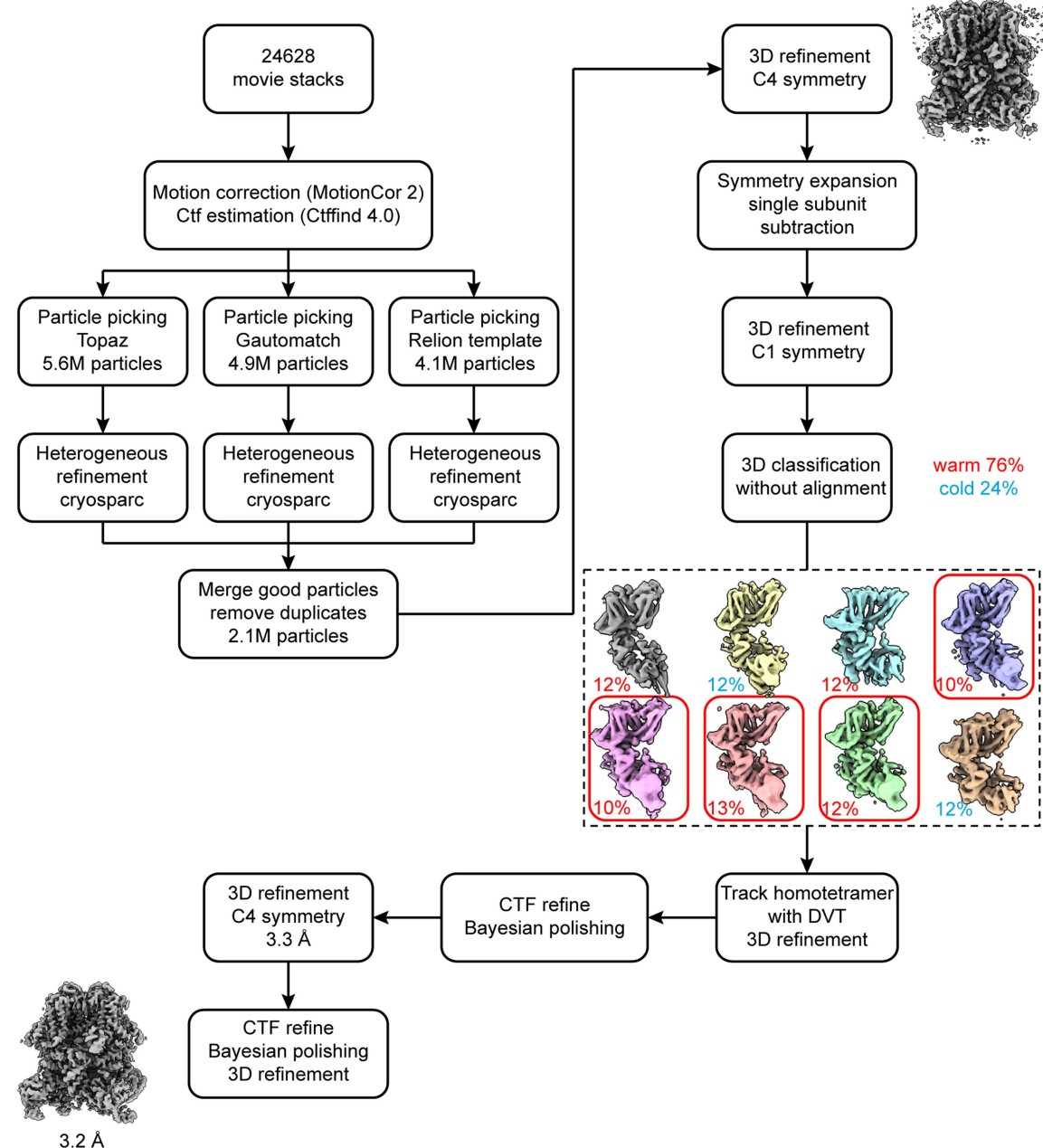

**Extended Data Fig. 4 | Cryo-EM data processing workflow of TRPM4 with 5 mM Ca²⁺ and 2 mM DVT at 37 °C.** Key maps are included. This figure also shows the percentage of the particles in warm and cold conformations at the subunit level.

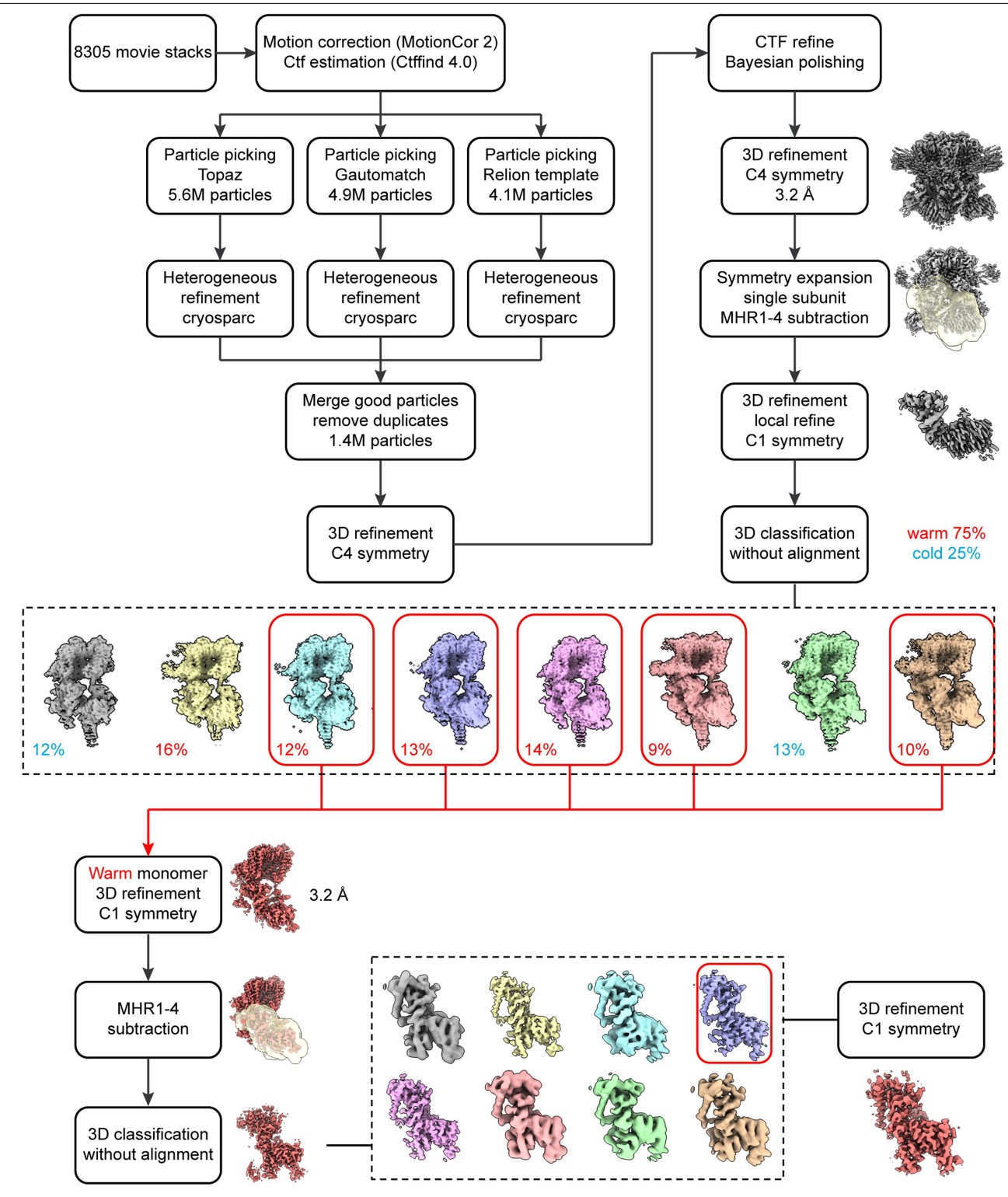

**Extended Data Fig. 5 | Cryo-EM data processing workflow of TRPM4 with 5 mM Ca²⁺ and 5 mM ATP at 37 °C.** Key maps and the mask (shown as a transparent envelope) used to subtract the intracellular domain (MHR1/2/3/4) are included. This figure also shows the percentage of the particles in warm and cold conformations at the subunit level.

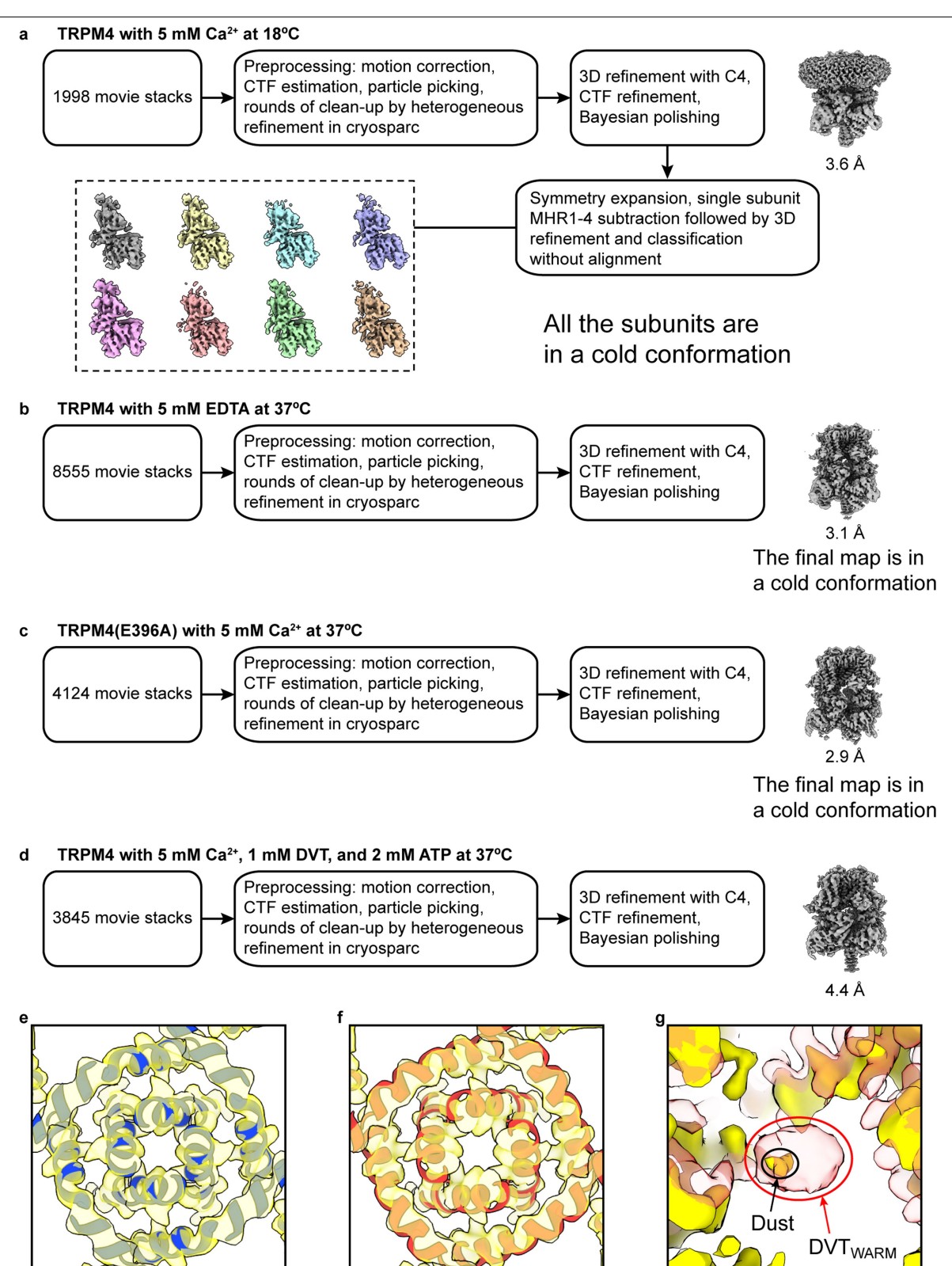

**Extended Data Fig. 6 | Cryo-EM data processing workflow of TRPM4 with 5 mM Ca²⁺ at 18 °C (a), TRPM4 with 5 mM EDTA at 37 °C (b), TRPM4(E396A) with 5 mM Ca²⁺ at 37 °C (c), and TRPM4 with 5 mM Ca²⁺, 1 mM DVT, and 2 mM ATP at 37 °C (d). e**, Superimposition of the Ca²⁺/DVT/ATP–TRPM4$_{WARM}$ cryo-EM map (yellow) and the Ca²⁺–TRPM4$_{WARM}$ model (blue; closed state), focusing on the transmembrane domain, viewed from the intracellular side. **f**, Superimposition of the Ca²⁺/DVT/ATP–TRPM4$_{WARM}$ cryo-EM map (yellow) and the Ca²⁺/DVT–TRPM4$_{WARM}$ model (salmon; open state), focusing on the transmembrane domain, viewed from the intracellular side. **g**, Superimposition of the TRPM4 cryo-EM maps of Ca²⁺/DVT/ATP–TRPM4$_{WARM}$ (yellow) and Ca²⁺/DVT–TRPM4$_{WARM}$ (salmon), focusing on the DVT$_{WARM}$ site and its surroundings.

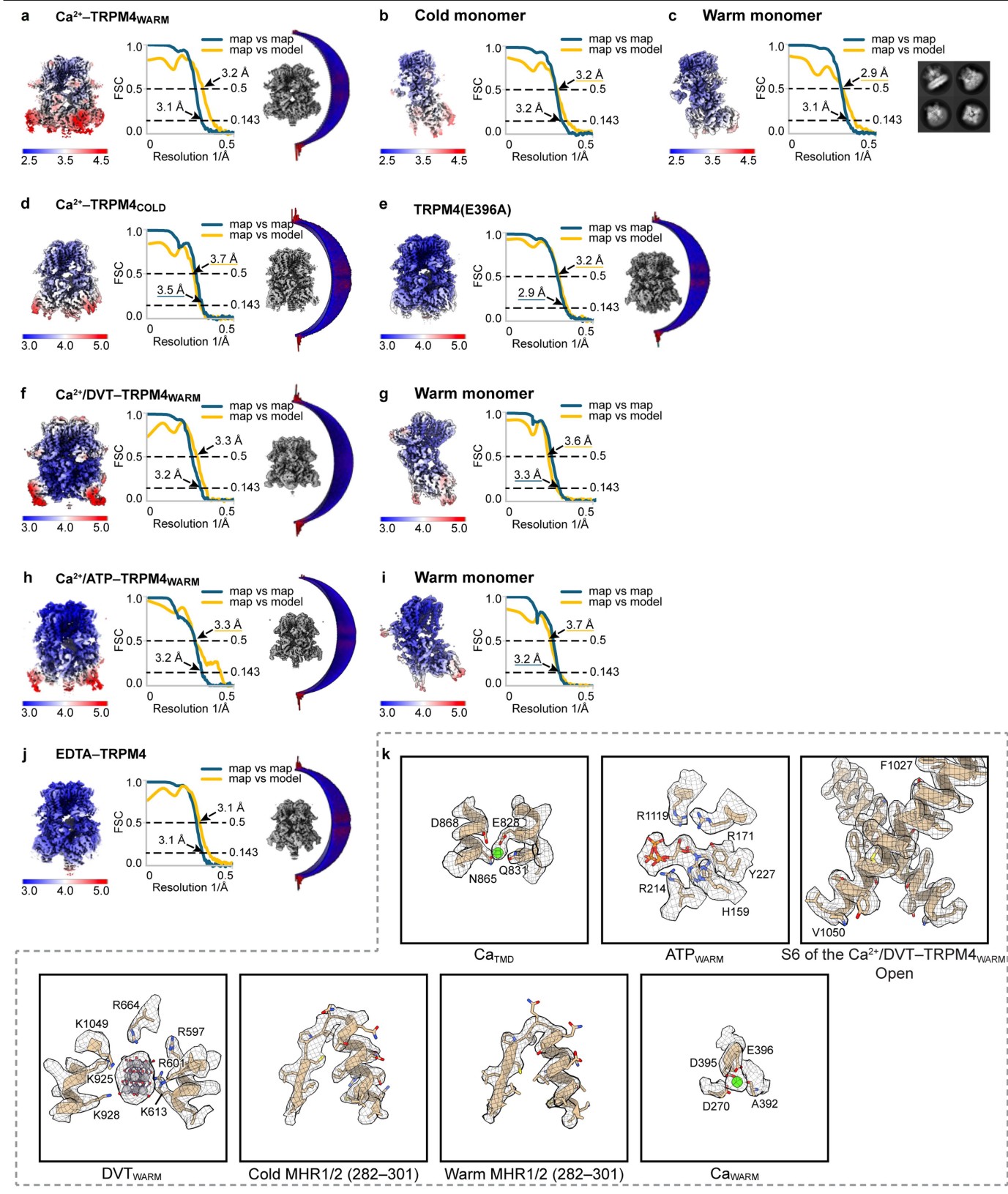

**Extended Data Fig. 7** | See next page for caption.

**Extended Data Fig. 7 | Cryo-EM analysis of the datasets in this paper. a**, From left to right, local resolution estimation, FSC curves for map vs. map (blue) and map vs model (orange), and angular distribution of the particles that give rise to the final cryo-EM map reconstruction for the $Ca^{2+}$–$TRPM4_{WARM}$ data. **b,c**, Local resolution estimation and FSC curves for the cold (**b**) and warm (**c**) monomer of the $Ca^{2+}$–$TRPM4_{WARM}$ data. The representative 2D class averages of the $Ca^{2+}$–$TRPM4_{WARM}$ data are shown on the far right of panel **c**. **d**–**g**, Local resolution estimation, FSC curves for map vs. map (blue) and map vs model (orange), and angular distribution of the particles that give rise to the final cryo-EM map reconstruction for $Ca^{2+}$–$TRPM4_{COLD}$ (**d**), $Ca^{2+}$–TRPM4(E396A) (**e**), EDTA–TRPM4 (**f**), $Ca^{2+}$/DVT–$TRPM4_{WARM}$ (**g**). **h**, Local resolution estimation and FSC curves for the warm monomer of the $Ca^{2+}$/DVT–$TRPM4_{WARM}$ data. **i**, Local resolution estimation, FSC curves for map vs. map (blue) and map vs model (orange), and angular distribution of the particles that give rise to the final cryo-EM map reconstruction for the $Ca^{2+}$/ATP–$TRPM4_{WARM}$ data. **j**, Local resolution estimation and FSC curves for the warm monomer of the $Ca^{2+}$/ATP–$TRPM4_{WARM}$ data. **k**, Representative densities.

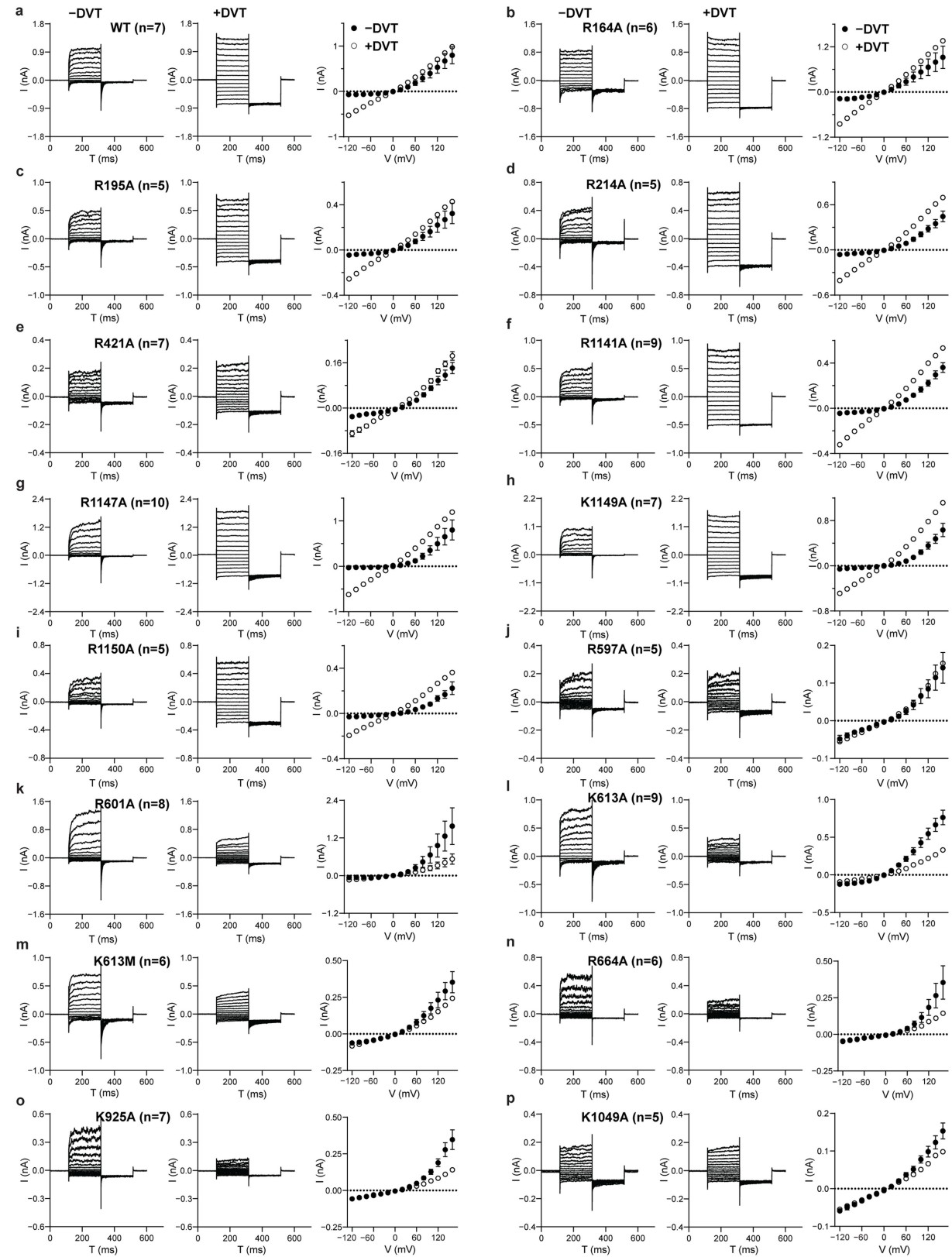

**Extended Data Fig. 8 | DVT modulation on wild-type TRPM4 and its variants at the DVT_COLD and DVT_WARM sites.** Representative current traces activated by 5 mM Ca²⁺ in the absence (left) or presence of 10 µM DVT (middle) from membrane patches excised from tsA201 cells overexpressing wild-type TRPM4 (**a**), variants at the DVTCOLD site (**b**–**i**), and variants in the DVTWARM site (**j**–**p**), recorded in the inside-out patch-clamp configuration. Voltage clamps were applied from 160 mV to −120 mV (200 ms each step) with a final tail pulse at −120 mV (200 ms), with a holding potential of 0 mV. The mean current amplitudes at the end of each pulse step were plotted as a function of clamp voltage (right). The number of patches is indicated in the parentheses after the construct name. Error bars represent SEM. Notably, adding DVT appears to inhibit the currents of certain mutants. While the exact mechanism behind these phenotypes remains unclear, it is possible that some mutations do not abolish DVT binding. Instead, these mutations may alter the interaction between DVT and the channel, thereby affecting the action of DVT on the channel.

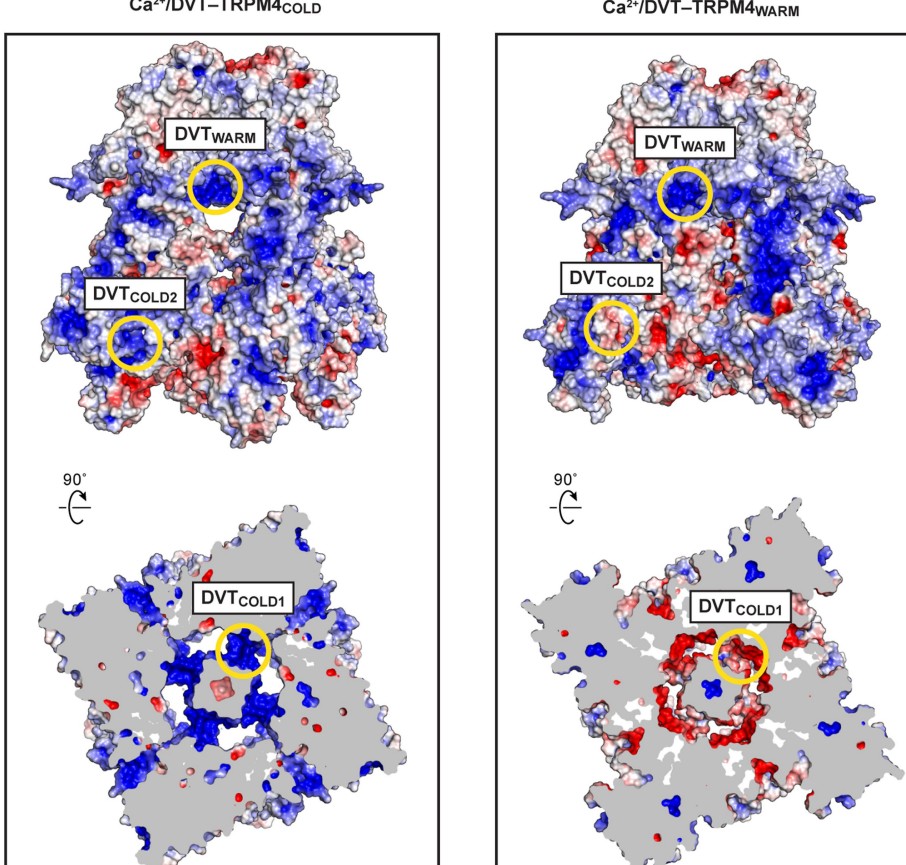

**Extended Data Fig. 9 | The surfaces of Ca²⁺/DVT–TRPM4_COLD (left) and Ca²⁺/DVT–TRPM4_WARM (right) coloured according to the electrostatic surface potential from −5 to 5 kT/e (red to blue).** The positions of the DVT_COLD and DVT_WARM sites are labelled.

**Extended Data Table 1 | Cryo-EM data collection, refinement and validation statistics**

| | Ca²⁺–TRPM4_WARM (EMDB-44360) (PDB 9B8W) | Ca²⁺–TRPM4_WARM Subunit (EMDB-44361) (PDB 9B8X) | Ca²⁺/DVT–TRPM4_WARM (EMDB-44362) (PDB 9B8Y) | Ca²⁺/DVT–TRPM4_WARM Subunit (EMDB-44363) (PDB 9B8Z) | Ca²⁺/ATP–TRPM4_WARM (EMDB-44364) (PDB 9B90) |
|---|---|---|---|---|---|
| **Data collection and processing** | | | | | |
| Magnification | 105000 | 105000 | 105000 | 105000 | 105000 |
| Voltage (kV) | 300 | 300 | 300 | 300 | 300 |
| Electron exposure (e–/Å²) | 50 | 50 | 50 | 50 | 50 |
| Defocus range (μm) | -1.2 – -1.9 | -1.2 – -1.9 | -1.2 – -1.9 | -1.2 – -1.9 | -1.2 – -1.9 |
| Pixel size (Å) | 0.828 | 0.828 | 0.828 | 0.828 | 0.828 |
| Symmetry imposed | C4 | C1 | C4 | C1 | C4 |
| Initial particle images (no.) | 1.4M | | 2.1M | | 1.4M |
| Final particle images (no.) | 990K | 389K | 113K | 243K | 991K |
| Map resolution (Å) | 3.1 | | 3.2 | 3.4 | 3. |
| FSC threshold | 0.143 | 0.143 | 0.143 | 0.143 | 0.143 |
| Map resolution range (Å) | 3.1 – 246.2 | 3.0 – 246.2 | 3.2 – 246.2 | 3.4 – 246.2 | 3.2 – 246.2 |
| **Refinement** | | | | | |
| Initial model used (PDB code) | 5WP6 | 5WP6 | 5WP6 | 5WP6 | 5WP6 |
| Model resolution (Å) | 3.2 | 3.5 | 3.3 | 3.6 | 3.3 |
| FSC threshold | 0.5 | 0.5 | 0.5 | 0.5 | 0.5 |
| Model resolution range (Å) | 3.2 – 246.2 | 3.5 – 246.2 | 3.3 – 246.2 | 3.6 – 246.2 | 3.3 – 246.2 |
| Map sharpening B factor (Å²) | -139.953 | -60 | -100.76 | -97.4044 | -138.547 |
| Model composition | | | | | |
| Non-hydrogen atoms | 30428 | 7596 | 30444 | 7469 | 30300 |
| Protein residues | 3988 | 997 | 3968 | 978 | 3988 |
| Ligands | 8 | 2 | 12 | 3 | 12 |
| B factors (Å²) | | | | | |
| Protein | 57.83 | 136.05 | 97.83 | 119.21 | 105.78 |
| Ligand | 74.76 | 138.43 | 181.47 | 130.97 | 122.15 |
| R.m.s. deviations | | | | | |
| Bond lengths (Å) | 0.006 | 0.002 | 0.003 | 0.003 | 0.003 |
| Bond angles (°) | 0.579 | 0.455 | 0.511 | 0.445 | 0.546 |
| Validation | | | | | |
| MolProbity score | 1.89 | 1.43 | 1.54 | 1.34 | 1.79 |
| Clashscore | 7.71 | 5.78 | 7.34 | 3.12 | 8.17 |
| Poor rotamers (%) | 2.16 | 0 | 0.23 | 0.67 | 0.20 |
| Ramachandran plot | | | | | |
| Favored (%) | 96.60 | 97.45 | 96.05 | 96.41 | 94.94 |
| Allowed (%) | 3.40 | 2.55 | 3.95 | 3.59 | 5.06 |
| Disallowed (%) | 0 | 0 | 0 | 0 | 0 |

This table contains information on the cryo-EM data collection and processing, as well as the refinement and validation of the atomic models.

**Extended Data Table 2 | Cryo-EM data collection, refinement and validation statistics (continued)**

| | Ca$^{2+}$/ATP–TRPM4$_{WARM}$ Subunit (EMDB-44365) (PDB 9B91) | Ca$^{2+}$–TRPM4$_{COLD}$ (EMDB-44366) (PDB 9B92) | EDTA–TRPM4 (EMDB-44367) (PDB 9B93) | Ca$^{2+}$TRPM4 (E396A) (EMDB-44368) (PDB 9B94) | Ca$^{2+}$/ATP–TRPM4$_{WARM}$ + DVT (EMDB-44369) |
|---|---|---|---|---|---|
| **Data collection and processing** | | | | | |
| Magnification | 105000 | 45000 | 105000 | 105000 | 45000 |
| Voltage (kV) | 300 | 200 | 300 | 300 | 200 |
| Electron exposure (e–/Å$^2$) | 50 | 60 | 50 | 50 | 60 |
| Defocus range (µm) | -1.2 – -1.9 | -1.4 – -2.2 | -1.2 – -1.9 | -1.2 – -1.9 | -1.4 – -2.2 |
| Pixel size (Å) | 0.828 | 1.16 | 0.828 | 0.828 | 1.16 |
| Symmetry imposed | C1 | C4 | C4 | C4 | C4 |
| Initial particle images (no.) | | 818K | 817K | 609K | 781K |
| Final particle images (no.) | 2.3M | 301K | 610K | 478K | 168K |
| Map resolution (Å) | 3.3 | 3.5 | 3.1 | 2.9 | 4.4 |
|   FSC threshold | 0.143 | 0.143 | 0.143 | 0.143 | 0.143 |
| Map resolution range (Å) | 3.3 – 246.2 | 3.5 – 246.2 | 3.1 – 246.2 | 2.9 – 246.2 | 4.4 – 246.2 |
| **Refinement** | | | | | |
| Initial model used (PDB code) | 5WP6 | 5WP6 | 5WP6 | 5WP6 | |
| Model resolution (Å) | 3.4 | 3.7 | 3.1 | 3.2 | |
|   FSC threshold | 0.5 | 0.5 | 0.5 | 0.5 | |
| Model resolution range (Å) | 3.4 – 246.2 | 3.7 – 246.2 | 3.1 – 246.2 | 3.2 – 246.2 | |
| Map sharpening $B$ factor (Å$^2$) | -138.021 | -138.927 | -117.774 | -47.6536 | |
| Model composition | | | | | |
|   Non-hydrogen atoms | 7478 | 30280 | 29920 | 29688 | |
|   Protein residues | 993 | 3976 | 3948 | 3880 | |
|   Ligands | 3 | 4 | 0 | 4 | |
| $B$ factors (Å$^2$) | | | | | |
|   Protein | 141.53 | 144.92 | 120.98 | 122.39 | |
|   Ligand | 205.18 | 126.84 | 0 | 99.43 | |
| R.m.s. deviations | | | | | |
|   Bond lengths (Å) | 0.004 | 0.003 | 0.002 | 0.004 | |
|   Bond angles (°) | 0.517 | 0.569 | 0.470 | 0.491 | |
| Validation | | | | | |
|   MolProbity score | 1.64 | 1.59 | 1.43 | 1.62 | |
|   Clashscore | 5.62 | 7.25 | 6.21 | 8.5 | |
|   Poor rotamers (%) | 0.2 | 0.13 | 0.07 | 0 | |
| Ramachandran plot | | | | | |
|   Favored (%) | 96.42 | 96.68 | 97.59 | 97.12 | |
|   Allowed (%) | 3.58 | 3.14 | 2.41 | 2.88 | |
|   Disallowed (%) | 0 | 0 | 0 | 0 | |

This table contains information on the cryo-EM data collection and processing, as well as the refinement and validation of the atomic models.

# Reporting Summary

## Statistics

For all statistical analyses, confirm that the following items are present in the figure legend, table legend, main text, or Methods section.

| n/a | Confirmed | |
|---|---|---|
| ☐ | ☒ | The exact sample size ($n$) for each experimental group/condition, given as a discrete number and unit of measurement |
| ☐ | ☒ | A statement on whether measurements were taken from distinct samples or whether the same sample was measured repeatedly |
| ☐ | ☒ | The statistical test(s) used AND whether they are one- or two-sided<br>*Only common tests should be described solely by name; describe more complex techniques in the Methods section.* |
| ☒ | ☐ | A description of all covariates tested |
| ☐ | ☒ | A description of any assumptions or corrections, such as tests of normality and adjustment for multiple comparisons |
| ☐ | ☒ | A full description of the statistical parameters including central tendency (e.g. means) or other basic estimates (e.g. regression coefficient) AND variation (e.g. standard deviation) or associated estimates of uncertainty (e.g. confidence intervals) |
| ☐ | ☒ | For null hypothesis testing, the test statistic (e.g. $F$, $t$, $r$) with confidence intervals, effect sizes, degrees of freedom and $P$ value noted<br>*Give P values as exact values whenever suitable.* |
| ☒ | ☐ | For Bayesian analysis, information on the choice of priors and Markov chain Monte Carlo settings |
| ☒ | ☐ | For hierarchical and complex designs, identification of the appropriate level for tests and full reporting of outcomes |
| ☒ | ☐ | Estimates of effect sizes (e.g. Cohen's $d$, Pearson's $r$), indicating how they were calculated |

*Our web collection on statistics for biologists contains articles on many of the points above.*

## Software and code

Policy information about availability of computer code

| Data collection | SerialEM 4.15, ClampFit 11.0.3 |
|---|---|
| Data analysis | Relion-4.0, Cryosparc-v2, Motioncor2-1.1.0, Ctffind-4.1.10, Gautomatch-0.56, Topaz-0.2.5, Coot-0.8.9.2, Phenix-1.20.1, UCSF chimeraX-0.91, Hole-2.2005, Pymol-2.3.2, ClamFit_11.0.3, GraphPad Prism 7, OriginPro 2024, eLBOW-1.20.1 |

For manuscripts utilizing custom algorithms or software that are central to the research but not yet described in published literature, software must be made available to editors and reviewers. We strongly encourage code deposition in a community repository (e.g. GitHub). See the Nature Portfolio guidelines for submitting code & software for further information.

## Data

Policy information about availability of data

All manuscripts must include a data availability statement. This statement should provide the following information, where applicable:
- Accession codes, unique identifiers, or web links for publicly available datasets
- A description of any restrictions on data availability
- For clinical datasets or third party data, please ensure that the statement adheres to our policy

Cryo-EM density maps have been deposited in the Electron Microscopy Data Bank (EMDB) under accession numbers EMD-44360 (Ca2+–TRPM4WARM), EMD-44361 (Ca2+–TRPM4WARM Subunit), EMD-44362 (Ca2+/DVT–TRPM4WARM), EMD-44363 (Ca2+/DVT–TRPM4WARM Subunit), EMD-44364 (Ca2+/ATP–TRPM4WARM), EMD-44365 (Ca2+/ATP–TRPM4WARM Subunit), EMD-44366 (Ca2+–TRPM4COLD), EMD-44367 (EDTA–TRPM4), EMD-44368 (Ca2+TRPM4(E396A))

and EMD-44369 (Ca2+/ATP–TRPM4WARM + DVT). Structure models have been deposited in the RCSB Protein Data Bank under accession codes 9B8W (Ca2+–TRPM4WARM), 9B8X (Ca2+–TRPM4WARM Subunit), 9B8Y (Ca2+/DVT–TRPM4WARM), 9B8Z (Ca2+/DVT–TRPM4WARM Subunit), 9B90 (Ca2+/ATP–TRPM4WARM), 9B91 (Ca2+/ATP–TRPM4WARM Subunit), 9B92 (Ca2+–TRPM4COLD), 9B93 (EDTA–TRPM4) and 9B94 (Ca2+TRPM4(E396A)).

# Research involving human participants, their data, or biological material

Policy information about studies with human participants or human data. See also policy information about sex, gender (identity/presentation), and sexual orientation and race, ethnicity and racism.

| Reporting on sex and gender | Does not apply |
|---|---|
| Reporting on race, ethnicity, or other socially relevant groupings | Does not apply |
| Population characteristics | Does not apply |
| Recruitment | Does not apply |
| Ethics oversight | Does not apply |

Note that full information on the approval of the study protocol must also be provided in the manuscript.

# Field-specific reporting

Please select the one below that is the best fit for your research. If you are not sure, read the appropriate sections before making your selection.

☒ Life sciences  ☐ Behavioural & social sciences  ☐ Ecological, evolutionary & environmental sciences

For a reference copy of the document with all sections, see nature.com/documents/nr-reporting-summary-flat.pdf

# Life sciences study design

All studies must disclose on these points even when the disclosure is negative.

| Sample size | The sample sizes of the cryo-EM data were not predetermined, but were determined/limited by the available time of the microscope. These sample sizes were large enough as they allowed for the reconstruction of cryo-EM maps of sufficiently high quality to draw the conclusions of this study. |
|---|---|
| Data exclusions | During cryo-EM data processing, particles that clearly did not represent the target protein or did not show high-resolution features were discarded. The data processing workflows are summarized in the Extended Data Figures. These workflows were predefined and represent common practices in cryo-EM image processing. |
| Replication | For the cryo-EM studies, initial data collection was performed on a 200 kV Arctica microscope. Subsequently, in order to validate the structural findings and to improve the quality of the data, larger datasets were collected on a 300 kV Titan Krios microscope. All attempts to replicate the structural findings were successful. For patch-clamp experiment, independent measurements/replicates are indicated in the figure legend. |
| Randomization | This study did not allocate experimental groups, therefore, no randomization was necessary. |
| Blinding | Blinding was not applicable to single-particle cryo-EM studies. |

# Reporting for specific materials, systems and methods

We require information from authors about some types of materials, experimental systems and methods used in many studies. Here, indicate whether each material, system or method listed is relevant to your study. If you are not sure if a list item applies to your research, read the appropriate section before selecting a response.

## Materials & experimental systems

| n/a | Involved in the study |
|---|---|
| ☒ | ☐ Antibodies |
| ☐ | ☒ Eukaryotic cell lines |
| ☒ | ☐ Palaeontology and archaeology |
| ☒ | ☐ Animals and other organisms |
| ☒ | ☐ Clinical data |
| ☒ | ☐ Dual use research of concern |
| ☒ | ☐ Plants |

## Methods

| n/a | Involved in the study |
|---|---|
| ☒ | ☐ ChIP-seq |
| ☒ | ☐ Flow cytometry |
| ☒ | ☐ MRI-based neuroimaging |

## Eukaryotic cell lines

Policy information about cell lines and Sex and Gender in Research

| | |
|---|---|
| Cell line source(s) | tsa201 cells from Sigma-Aldrich |
| Authentication | Cells are obtained from the vendor and is not independently authenticated |
| Mycoplasma contamination | Cells are tested free from Mycoplasma contamination |
| Commonly misidentified lines (See ICLAC register) | None |

## Plants

| | |
|---|---|
| Seed stocks | Does not apply |
| Novel plant genotypes | Does not apply |
| Authentication | Does not apply |

