## [Peer Review File · Nature]

Manuscript Title: Physiological Temperature Drives TRPM4 Ligand Recognition and Channel Gating

Reviewer Comments & Author Rebuttals

Reviewer Reports on the Initial Version:

Referees' comments:

Referee #1 (Remarks to the Author):

In this manuscript, the authors examine structures of a thermally sensitive TRP channel (TRPM4) held at a physiologically relevant temperature of 37 degrees C prior to rapid freezing, This is compared to structural analysis for TRPM4 protein held at a lower., more typical temperature of 4-18 degrees C. The motivation is to address the idea that protein structure needs to be assessed under conditions that most closely approximate physiological state - including thermal state.

The authors have chosen an interesting model for this approach by focusing on a member of the thermosensitive TRP channel family, namely TRPM4. While TRPM4 is not a temperature sensor per se, it's activity does show pronounced temperature dependence, making it a relevant model for emphasizing the importance of determining structures, when possible, at relevant temperatures. Perhaps the most compelling finding in support of this message is that the 'warm' conformation observed in this study exhibits binding sites for factors or modulators (e.g., calcium and ATP) that are not seen in the 'cold' conformation. Overall, the study is interesting and well performed. The results are convincing and the message will be widely appreciated by the cryo-EM field.

The authors may wish to consider the following minor suggestions:

1. while TRPM4 exhibits a degree of thermosensitivity, at least in vitro, that is a characteristic of certain other members of this ion channel family, it is not a canonical temperature-activated channel in the physiologic sense. While the real message from this study is that protein structures should ideally be examined at their physiologically relevant temperature, the message is a bit confused by the use of the terms 'warm' and 'cold' conformations, which may imply to some readers that this pertains to TRPM4 as

a temperature modulated channel in vivo. This could potentially distract from the main message of the study.

2. while the authors do cite papers from the Lee and Sobolevsky (refs 63 and 64), these papers should really be cited early in the manuscript, perhaps in the introduction, as they also illustrate the same goal of examining cryo-EM structures for proteins incubated at higher, physiologically relevant temperatures prior to freezing.

Referee #2 (Remarks to the Author):

In this study, Hu et al. observed significant conformational changes in TRPM4 induced by increasing the temperature to 37°C. These changes led to the emergence of new ligand-binding sites in a conformation that was previously unobserved. Using electrophysiology, they demonstrated that the binding of ligands, such as Ca²⁺, DVT, and ATP, to these new sites is functionally relevant. The experiments were conducted with robustness, and the data quality is impressive. The work provides an alternative mechanism for temperature sensing that is distinct from the TRPV channel, thus offering new insights into the gating of temperature-sensitive TRP channels. The work also highlights the importance of studying temperature-sensitive proteins at relevant temperatures, which is often overlooked. I found this study very interesting and believe it is a good fit for Nature. I recommend its publication, provided the following questions are discussed.

1. Line 30: I have no doubts about Ca and DVT, but I am not entirely convinced that one can say ATP binds to a 'different' site. From the structure, it seems to me that it is the ATP-binding domain (MHR1/2) that moves, so the position where ATP binds relative to the channel changes. As the author states in lines 263-264 and line 272, the majority of the ATP site remains unchanged compared to the cold structure. It is just that the ATP tail is now exposed. Moreover, in the warm structure, the ATP tail is in fact not very close to the rib helix. It is uncertain whether they form meaningful interactions. For these reasons, I am not sure this justifies calling the warm ATP site a 'different' site.

Have the author considered another interpretation: that the movement of the MHR1/2 breaks part of the ATP site, so ATP inhibition at 37°C is weaker?

2. Lines 60-62: It reads as if the authors are the first to reveal that macromolecules can have these properties. However, this phenomenon has been described before in other systems (for example, doi: 10.7554/eLife.84632). I recommend removing the two words "macromolecules like" in the sentence to be more accurate.

3. Line 68: Did the author record currents at different temperatures from the same cell? Otherwise, it is not accurate to state that 37°C increases currents threefold. Variations in the expression level of channels across different cells can easily account for the observed effect.

4. Lines 188-189: It may not be accurate to state that channel opening requires depolarization. From Extended Data Figure 1, it seems that at hyperpolarization to -100 mV, the channel is still open. The channel may desensitize more at -100 mV, but it does not require depolarization to open.

5. Lines 200-201: Could the author elaborate more on why DVT binding does not shift the equilibrium? Is it merely a coincidence that the ratio is similar? One might imagine that the equilibrium would change because DVT binds to different sites under cold and warm conditions, and more importantly, the warm DVT-bound state is an open state, not a closed one. If there are three interchangeable states: cold closed, warm closed, and warm open, shouldn't DVT binding shift the channel from warm closed to warm open, thereby dragging the entire distribution toward warm open and shifting the equilibrium?

6. Lines 239-242: These statements are not correct. For voltage-gated K⁺, Na⁺, and Ca²⁺ channels, it is primarily S4 that moves, not necessarily the entire S1-S4. Additionally, 1/4 of a helical turn does not resemble most of the S4 movement in voltage-gated K⁺, Na⁺, and Ca²⁺ channels.

7. Lines 271-273: The distance between the ATP tail and R1119 or K1115 on the rib helix doesn't seem to be closer than the distance between the ATP tail and R422 in the cold structure. Could the author elaborate on why the tail in the warm structure is "favorably" surrounded compared to the cold structure?

8. Extended Data Figure 10: In the K to P panels, adding DVT appears to inhibit the channel in these mutants. Could the authors provide any hypotheses?

Author Rebuttals to Initial Comments:

Referees' comments:

Referee #1 (Remarks to the Author):

In this manuscript, the authors examine structures of a thermally sensitive TRP channel (TRPM4) held at a physiologically relevant temperature of 37 degrees C prior to rapid freezing, This is compared to structural analysis for TRPM4 protein held at a lower., more typical temperature of 4-18 degrees C. The motivation is to address the idea that protein structure needs to be assessed under conditions that most closely approximate physiological state - including thermal state.

The authors have chosen an interesting model for this approach by focusing on a member of the thermosensitive TRP channel family, namely TRPM4. While TRPM4 is not a temperature sensor per se, it's activity does show pronounced temperature dependence, making it a relevant model for emphasizing the importance of determining structures, when possible, at relevant temperatures. Perhaps the most compelling finding in support of this message is that the 'warm' conformation observed in this study exhibits binding sites for factors or modulators (e.g., calcium and ATP) that are not seen in the 'cold' conformation. Overall, the study is interesting and well performed. The results are convincing and the message will be widely appreciated by the cryo-EM field.

The authors may wish to consider the following minor suggestions:

1. while TRPM4 exhibits a degree of thermosensitivity, at least in vitro, that is a characteristic of certain other members of this ion channel family, it is not a canonical temperature-activated channel in the physiologic sense. While the real message from this study is that protein structures should ideally be examined at their physiologically relevant temperature, the message is a bit confused by the use of the terms 'warm' and 'cold' conformations, which may imply to some readers that this pertains to TRPM4 as a temperature modulated channel in vivo. This could potentially distract from the main message of the study.

Respond. We agree with the reviewer and to avoid potential confusion, we have revised the text to: "We refer to these conformations as "cold" and "warm", reflecting the temperatures at which they were identified and for ease of discussion. As discussed later, we speculate that both conformations may co-exist at physiological temperatures, with their equilibrium controlled by local Ca²⁺ concentration."

2. while the authors do cite papers from the Lee and Sobolevsky (refs 63 and 64), these papers should really be cited early in the manuscript, perhaps in the introduction, as they also illustrate the same goal of examining cryo-EM structures for proteins incubated at higher, physiologically relevant temperatures prior to freezing.

Respond. Thanks for the suggestion. We have moved this part to the introduction, lines 57 to 60.

Referee #2 (Remarks to the Author):

In this study, Hu et al. observed significant conformational changes in TRPM4 induced by increasing the temperature to 37°C. These changes led to the emergence of new ligand-binding sites in a conformation that was previously unobserved. Using electrophysiology, they demonstrated that the binding of ligands, such as Ca²⁺, DVT, and ATP, to these new sites is functionally relevant. The experiments were conducted with robustness, and the data quality is impressive. The work provides an alternative mechanism for temperature sensing that is distinct from the TRPV channel, thus offering new insights into the gating of temperature-sensitive TRP channels. The work also highlights the importance of studying temperature-sensitive proteins at relevant temperatures, which is often overlooked. I found this study very interesting and believe it is a good fit for Nature. I recommend its publication, provided the following questions are discussed.

1. Line 30: I have no doubts about Ca and DVT, but I am not entirely convinced that one can say ATP binds to a 'different' site. From the structure, it seems to me that it is the ATP-binding domain (MHR1/2) that moves, so the position where ATP binds relative to the channel changes. As the author states in lines 263-264 and line 272, the majority of the ATP site remains unchanged compared to the cold structure. It is just that the ATP tail is now exposed. Moreover, in the warm structure, the ATP tail is in fact not very close to the rib helix. It is uncertain whether they form meaningful interactions. For these reasons, I am not sure this justifies calling the warm ATP site a 'different' site.

Have the author considered another interpretation: that the movement of the MHR1/2 breaks part of the ATP site, so ATP inhibition at 37°C is weaker?

Respond. We appreciate this insightful comment. It is indeed possible that the MHR1/2 movement breaks part of the ATP site, potentially resulting in weaker ATP inhibition at 37°C. We have addressed this point in lines 273 to 276.

We also agree that a major part of the ATP site (the adenosine cleft) remains largely unchanged between the TRPM4 cold and warm conformations. However, the spatial locations of ATP_{COLD} and ATP_{WARM} are distinct, and as a result, whereas the ATP_{COLD} site engages two adjacent subunits, the ATP_{WARM} site involves only one subunit. This subtle variation in coordinating residues, coupled with the large spatial difference, may influence ATP's inhibitory potency at different temperatures, as suggested by the reviewer. Following the reviewer's suggestion, we have now referred to ATP_{WARM} as a different location instead of different binding site.

2. Lines 60-62: It reads as if the authors are the first to reveal that macromolecules can have these properties. However, this phenomenon has been described before in other systems (for example, doi: 10.7554/eLife.84632). I recommend removing the two words "macromolecules like" in the sentence to be more accurate.

Respond. We have followed the reviewer's suggestion and removed "macromolecules like".

3. Line 68: Did the author record currents at different temperatures from the same cell? Otherwise, it is not accurate to state that 37°C increases currents threefold. Variations in the expression level of channels across different cells can easily account for the observed effect.

Respond. These recordings were from different cells. We used the same protocol for transient transfection, and ensured similar expression times prior to recordings. We patched a relatively large number of cells (16 and 15 cells at 22°C and 37°C, respectively). The results consistently showed an increase in current at 37°C compared to 22°C, with an approximate threefold increase of the average value (Fig. 3e). We acknowledge that variations in the protein expression level may still be significant despite our efforts, and have revised the text to: "a substantial increase".

4. Lines 188-189: It may not be accurate to state that channel opening requires depolarization. From Extended Data Figure 1, it seems that at hyperpolarization to -100 mV, the channel is still open. The channel may desensitize more at -100 mV, but it does not require depolarization to open.

Respond. We agree with the reviewer that it is not accurate to state channel opening requires depolarization after desensitization. Our intention with the original statement was to emphasize that the open probability of the channel appears low at 0 mV. We have now revised the text to: "indicating a low open probability at zero or negative potentials".

5. Lines 200-201: Could the author elaborate more on why DVT binding does not shift the equilibrium? Is it merely a coincidence that the ratio is similar? One might imagine that the equilibrium would change because DVT binds to different sites under cold and warm conditions, and more importantly, the warm DVT-bound state is an open state, not a closed one. If there are three interchangeable states: cold closed, warm closed, and warm open, shouldn't DVT binding shift the channel from warm closed to warm open, thereby dragging the entire distribution toward warm open and shifting the equilibrium?

Respond. This is an interesting point. Our findings indicate that physiological temperature and Ca_{WARM} binding play a primary role in overcoming the energy barrier between the cold and warm (both closed and open combined) conformations. This transition involves breaking and creating intersubunit interfaces. On the other hand, DVT_{WARM} binding seems to primarily affect the equilibrium between the warm closed and warm open states—a process that involves relatively small conformational changes—but may be insufficient to impact the broader equilibrium between the cold and warm conformations.

It may also be a coincidence, as suggested by the reviewer, since DVT also binds to the cold conformation. In other words, the influence of DVT_{COLD} and DVT_{WARM} binding may counteract each other, resulting in a similar overall equilibrium.

6. Lines 239-242: These statements are not correct. For voltage-gated K⁺, Na⁺, and Ca²⁺ channels, it is primarily S4 that moves, not necessarily the entire S1-S4. Additionally, 1/4 of a helical turn does not resemble most of the S4 movement in voltage-gated K⁺, Na⁺, and Ca²⁺ channels.

Respond. Thanks for pointing this out. We have removed this statement.

7. Lines 271-273: The distance between the ATP tail and R1119 or K1115 on the rib helix doesn't seem to be closer than the distance between the ATP tail and R422 in the cold structure. Could the author elaborate on why the tail in the warm structure is "favorably" surrounded compared to the cold structure?

Respond. The reviewer is correct that the phosphate tail of ATP_{WARM} only loosely engages the positively charged residues on the rib helix. Our intention was to convey that these positive charges provide a favorable environment to accommodate the negatively charged ATP tail, rather than implying that this environment is more favorable environment than of R422 in the cold conformation. To avoid potential confusion, we have now removed the word "favorably".

8. Extended Data Figure 10: In the K to P panels, adding DVT appears to inhibit the channel in these mutants. Could the authors provide any hypotheses?

Respond. This is an interesting point. The DVT_{WARM} site is formed by seven positively charged residues from the pore domain, TRP helix, and S5-S6 linker. These structural elements are crucial for channel gating and are sensitive to alternations caused by mutation or ligand binding. While we do not fully understand the exact mechanism of this phenotype, it's conceivable that mutating some of the positively charged residues may not abolish DVT binding. Instead, these mutations may alter the interaction between DVT and the channel, thereby affecting the action of DVT on the channel. We have now discussed this point in the figure legend of ED Fig. 10, as there is not enough space to include it in the main text.

Reviewer Reports on the First Revision:

Referees' comments:

Referee #2 (Remarks to the Author):

The authors have addressed all my questions satisfactorily. I recommend the publication of this important work.